# Analysis of NS2-dependent effects on influenza PB1 segment extends replication requirements beyond the canonical promoter

Sharmada Swaminath [1,3], Marisa Mendes [1,3], Yipeng Zhang [1], Kaleigh A. Remick[2], Isabel Mejia [1], Melissa Güereca[1], Aartjan J. W. te Velthuis [2] & Alistair B. Russell [1] ✉

Influenza A virus encodes conserved promoter sequences. Using minimal replication assays—transfections with viral polymerase, nucleoprotein, and a genomic template—these sequences were identified as 13nt at the 5' end of the genomic RNA (U13) and 12nt at the 3' end (U12). Other than the fourth 3' nucleotide, the U12 and U13 sequences are identical between all eight RNA molecules of the segmented influenza A genome. However, individual segments can exhibit different dynamics during infection. Influenza NS2, which modulates transcription and replication differentially between genomic segments, may provide an explanation. Here, we assess how internal sequences of two genomic segments, HA and PB1, contribute to NS2-dependent replication and map such interactions down to individual nucleotides in PB1. We find that the expression of NS2 significantly alters sequence requirements for efficient replication beyond the identical U12 and U13 sequences, providing a potential mechanism for segment-specific replication dynamics across the influenza genome.

RNA viruses are constrained in genome size as a result of limitations from packaging and mutational burden[1]. Due to these constraints it is quite common for viral genomes to encode multifunctional proteins to maximize their use of limited genomic real estate.

Influenza A virus (IAV) is a negative-sense, segmented, single-stranded RNA virus. IAV's eighth, and smallest, genomic segment encodes two multifunctional non-structural proteins, NS1 and NS2. NS2 is also referred to as nuclear export protein (NEP) due to its role in facilitating genome egress from the nucleus to the cytoplasm[2,3]. This protein also aids in other functions such as viral replication and budding[4–10].

Once IAV viral ribonucleoproteins (vRNPs) are imported into the nucleus of an infected host cell, the viral polymerase at the terminus of each vRNP interacts with host RNA polymerase II to cap-snatch and cleave a nascent transcript to act as a primer for positive-sense viral mRNA transcription[11–14]. After viral mRNAs are translated, polymerase proteins are shuttled into the nucleus and initiate viral replication. These additional viral polymerases use the viral genomes (vRNA) as a template to synthesize positive-sense complementary RNA (cRNA), which in turn becomes a template for further production of vRNAs[15,16].

NS2 has been shown to interact with the heterotrimeric viral polymerase and alter the ratio of vRNA, cRNA, and mRNA in minimal replication (also called minireplicon) assays—coordinating an increase in cRNA synthesis and decrease in mRNA transcription[4,5,8,17]. Additionally, polymorphisms in NS2 can lead to variation in the frequency

[1]Department of Molecular Biology, School of Biological Sciences, University of California, San Diego, 9500 Gilman Drive, La Jolla, CA, USA. [2]Lewis Thomas Laboratory, Department of Molecular Biology, Princeton University, Princeton, NJ, USA. [3]These authors contributed equally: Sharmada Swaminath, Marisa Mendes. ✉e-mail: a5russell@ucsd.edu

of defective viral genome accumulation, further supporting its role in controlling genome replication[18,19]. Lastly, NS2 influences the response of avian influenza polymerases to ANP32A, a host dependency factor and key determinant of species tropism of avian IAV[20–22].

Using a combination of length-variant libraries in HA and PB1, and sequence-variant libraries in PB1, we now establish that NS2 modulates IAV replication to a greater degree than perhaps previously appreciated.

Most critically, we now know that the canonical viral promoters, considered to be both essential and sufficient for genome replication, cannot by themselves fully explain the dynamics of replication in the presence of NS2[23–25]. Specifically, in our sequence-variant libraries in PB1 we identify sites that are required for efficient viral replication in the presence of NS2 which are not conserved between IAV's genomic segments, in contrast with the canonical viral promoter. As these sites differ between segments, they may explain how IAV fine-tunes replication between its genomic segments, which exhibit different dynamics despite sharing identical core promoter sequences[26–30].

## Results

### Divergent length selection in minireplicons and infections

Since the original development of reverse genetics there have been numerous studies of selection pressures on the genome sequence of IAV. These pressures include both broad, architectural, selection on elements that are constrained by polymerase processivity and/or nucleoprotein coating, as well as site-specific constraints such as a need to retain the minimal viral promoter and vRNP bundling sequences[31–36]. These steps in the life cycle shape the fitness of any given viral genome and place fundamental limits on viral evolution.

These selection pressures exist alongside, and potentially in conflict with, a need to encode functional protein products. This makes it difficult to study selection pressures on genomic sequence independent of protein coding—does a mutation lead to loss of fitness because it changes amino acid sequence, or because it influences genome replication or packaging? However, this difficulty only strictly applies to replication-competent viruses. Viral populations also contain defective interfering particles (DIPs), which encode a non-standard, or defective, viral genome (nsVG, DVG)[37,38]. For IAV these genomes generally (but not exclusively) contain large internal deletions in at least one of three genomic segments encoding the heterotrimeric polymerase, PB2, PB1, and PA[39–42]. For our purposes, DVGs, including deletions as observed in IAV, reduce or remove selection on protein coding (other than the encoding of detrimental products), while retaining selection during genome replication and packaging[43]. Thus these mutations can reveal selection pressures that are otherwise masked.

In a previous publication from our group (Mendes and Russell[36]), we generated artificial length-variant barcoded, DVG vRNA libraries in the HA (haemagglutinin) and PB1 (polymerase basic 1) segments of the IAV strain A/WSN/1933[36]. In that study, we transfected these libraries alongside plasmids encoding the minimal viral replication machinery—PB2, PB1, PA, and NP—and tracked how length of a vRNA influences replication. This work revealed that smaller variants were able to replicate much faster than their longer counterparts, and that this selection was balanced with selection for longer fragments during packaging (Fig. 1a, b). Therefore, at least for the lengths we tested, polymerase processivity appears to be the key rate-limiting step in influenza genome replication. In contradiction to this simple model, a concurrent publication, Alnaji et al., found that in a single-round viral infection competition assay, a 395nt variant of the PB2 segment was outcompeted by a full-length counterpart during genome replication[44].

This discrepancy might be due to elements present during viral infection that are not included in minimal replication assays, or, alternatively, could be due to some effect specific to the variant studied in Alnaji et al. With respect to the latter hypothesis, it has been observed that different deletions may influence polymerase processivity through the formation of alternative RNA structures, rather than, or in addition to, length-dependent effects[45,46]. To explore further, using the same virus as our prior study (A/WSN/1933) we generated barcoded variants of PB1 and HA of 200, 400, 800, and 1600nt. We then rescued these variants using infectious virus to complement missing components, and performed single-

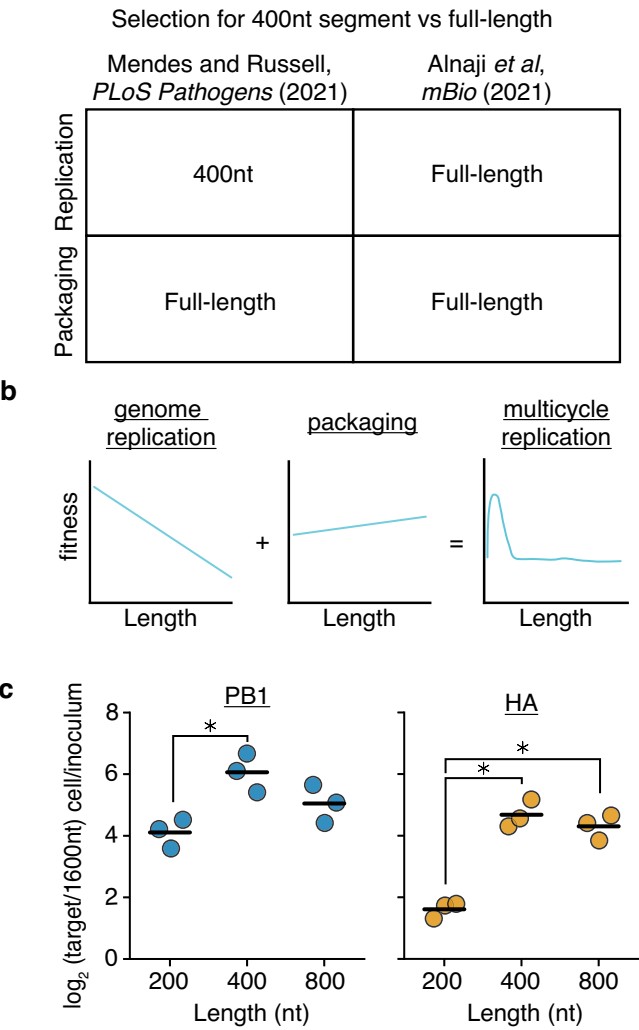

**Fig. 1 | Prior models incompletely describe genome replication during viral infection. a** Our previous work found that 400nt variants of the PB1 and HA segments outcompete their full-length counterparts during genome replication, but during packaging, full-length variants have an advantage. More comprehensive library-based analyzes of genomic length showed that smaller sizes correlate with better replication efficiency and longer sizes better packaging efficiency. Inconsistent with this model, Alnaji et al. described a phenomenon where a short variant of the PB2 segment was outcompeted during genome replication. **b** A schematic describing our model derived from measuring thousands of length-variants in Mendes and Russell[36]. **c** RNA genomes of barcoded 200, 400, 800, and 1600nt variants of A/WSN/1933 PB1 and HA bearing equal sequence length from 5' and 3' ends were generated from co-transfected plasmids using minimal replication machinery and rescued into virions by coinfection with wild-type virus. Viral supernatant was used to infect A549 cells at an MOI of 25, and qPCR was used to analyze the proportion of each variant before infection and within infected cells at 8 hours post-infection. 200, 400, and 800nt variants are assessed by their frequency relative to the 1600nt species. Asterisks indicate significantly different values, ANOVA $p < 0.05$, with post-hoc Tukey test, $q < 0.05$. $n = 3$, individual replicates and mean shown. Source data are provided as a Source Data file.

round infection competition assays at an MOI of 25. Similar to the results from Alnaji et al., and in contradiction to our simplistic model, a 400nt variant outcompeted its 200nt counterpart during a viral infection (Fig. 1c). While the 1600nt variant we tested continued to be the least fit length, the fitness curve in this experiment indicates that there is not a straightforward, linear relationship between segment replication and length. Nevertheless, IAV DVGs continue to exhibit a selective advantage during replication due to their reduced length, but in a more complex nature than we had initially appreciated.

## NS2 expression alters length-dependent replication

The differences in length-dependent replication dynamics between infection and minimal replication assays suggests that there is at least one missing factor in our experiments. It has been reported that NS2 can regulate the switch from viral transcription to replication, and so

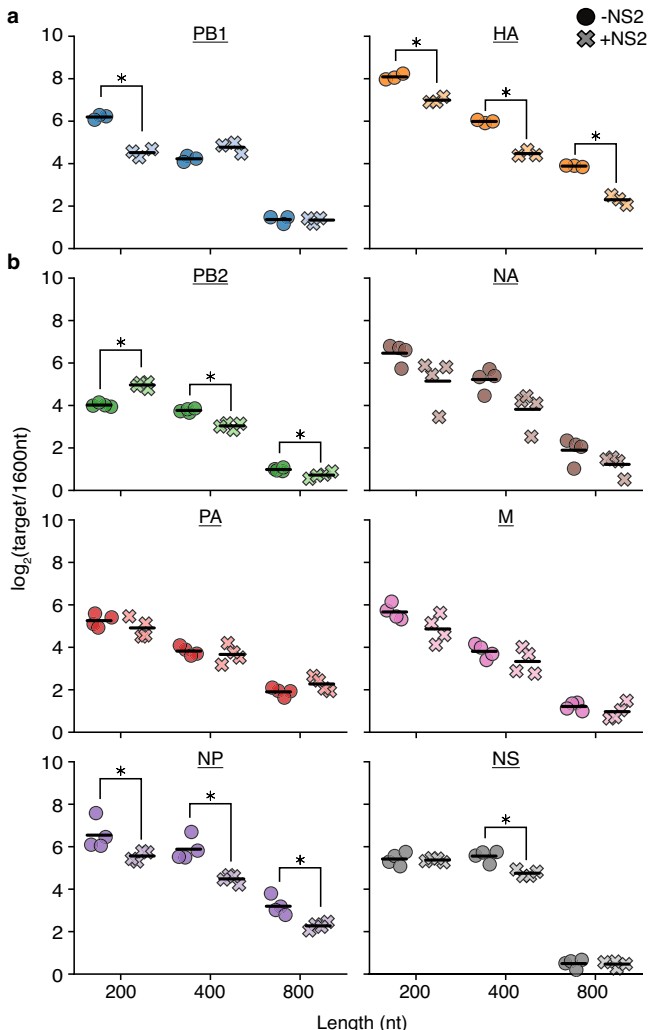

**Fig. 2 | Expression of NS2 complicates simple length-dependent replication kinetics. a**, **b** Barcoded 200, 400, 800, and 1600nt variants of each individual segment bearing equal sequence length derived from 5′ and 3′ ends and lacking canonical start codons were generated and co-transfected into HEK293T cells along with the minimal replication machinery with, or without, NS2. The relative frequency of vRNA of each variant was measured by qPCR at 24 hours post-transfection, with 1600nt variants serving as a comparison. Asterisks indicate conditions significantly impacted by the expression of NS2, two-sample two-tailed t-test with a within-panel Benjamini-Hochberg corrected FDR < 0.05. $n = 3$ for (**a**), $n = 4$ for (**b**), individual replicates and mean displayed. Source data are provided as a Source Data file.

this may contribute to our observations[4–7,17]. To investigate further, we used our PB1 and HA barcoded variants in a minimal replication assay with, and without, the additional expression of NS2 (Fig. 2a). In these experiments, and all other qPCR presented, we use a vRNA-specific primer to convert RNA to cDNA. From the work of the Kawaoka group, this should largely sample vRNA, with ~3-5% contamination from mRNA and cRNA due to mispriming or primer-independent conversion of RNA to cDNA[27]. We would also like to stress that we perform these, and many other qPCR experiments throughout this work as replication competition experiments, where multiple templates are measured within a single transfection experiment. As presented in Supplementary Fig. 1, this type of experiment can capture kinetic differences that may not otherwise result in changes to final, saturated, steady-state values. We deliberately use these assays to explore influenza genome replication as it has been previously observed that flu segments compete with one-another for some rate-limiting element in minimal replication assays[47]. This observation demonstrates that changes to synthesis rate alone may only influence steady-state concentrations in a subset of examples that overcome a requirement for some other rate-limiting process, and that steady-state, single template, assays may under-measure how variation in the genome alters replication kinetics. We note that such assays do have their drawbacks, such as inappropriate sampling of *trans*-acting selection pressures, but as throughout this work we are generally interested in *cis* effects we believe this to be a reasonable compromise.

When we perform these experiments we see that while HA and PB1 replicate similarly in the absence of NS2, they differ in their response to NS2 expression. For PB1, we observe a dramatic decrease in replication of a 200nt variant relative to a 1600nt control, whereas for HA we see that all lengths tested lose some of their advantage over that same control. This effect was not unique to NS2 from A/WSN/1933, as when we repeat these experiments with NS2 from a different lab adapted H1N1 (A/PR/8/1934), a circulating H1N1 (A/Cal/07/2009), or a circulating H3N2 (A/Sydney/05/1997), we observe similar results (Supplementary Fig. 2).

The different effects between HA and PB1 prompted us to explore further. Although NS2 appears to universally promote cRNA synthesis and decrease mRNA synthesis across the eight genomic segments of IAV, the precise magnitude of these effects appears to differ between each in minimal replication assays[4,5]. We therefore performed similar measurements across the remaining six segments (Fig. 2b). While some of these segments, such as NP, exhibited behavior similar to HA, other segments displayed trends inconsistent with either PB1 or HA. For instance, the shortest length in PB2 showed a replication advantage upon NS2 expression. Lastly, any data point showing reduced (or increased) replication relative to a 1600nt variant is technically explainable by a change in replication of the 1600nt control alone. Comparing abundance of the 1600nt variant in these same assays against a housekeeping control, we find that expression of NS2 does not impact replication of this template, and so the differences we observe in our experiments are likely the results of NS2 expression suppressing, or enhancing, replication of the indicated variants (Supplementary Fig. 3). It is unlikely that our results can be explained by polymerase processivity alone because we would anticipate increased processivity on NS2 expression to uniformly favor larger variants across all segments, which we do not observe.

## Analysis of region-specific selection under NS2 expression

Ultimately the impact of NS2 expression in minimal replication assays was highly idiosyncratic, producing a spectrum of effects between different influenza segments. One possibility for this seeming lack of pattern is that the particular junctions we chose may, or may not, be representative of the general behavior of that segment. To address this concern, and obtain more comprehensive measurements on the effect of NS2 expression on replication, we returned to our existing length-

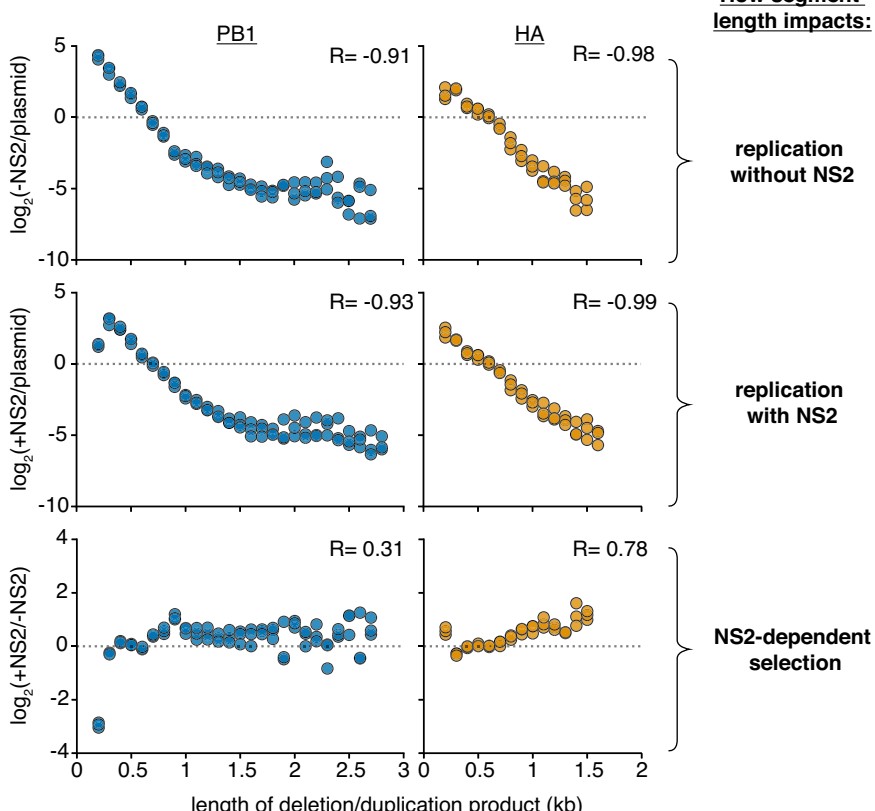

**Fig. 3 | Analysis of length variant libraries confirms NS2-dependent replication effects.** Size distributions of libraries in transfections with minimal replication machinery only (−NS2) and with the addition of NS2 (+NS2) 24 hours post-transfection in HEK293T cells as compared to the original plasmid library or one another. The fraction of variants falling within each 100nt bin was compared. Points above the dotted line represent sizes in each individual library which were enriched, below, depleted. Points are only shown if represented in all three libraries under both conditions. R is the Spearman correlation coefficient. $n = 3$, all replicates shown. Inter-replicate correlation plots presented in Supplementary Fig. 4. Source data are provided as a Source Data file.

variant libraries in PB1 and HA[36]. In brief, these libraries consist of ~1900 PB1, and ~850 HA, variants. In constructing these libraries, we randomly assemble a 5' region of vRNA with a 3' region and a 12nt barcode, leading to both deletions and duplications which we can track by simply sequencing the associated barcode. Every region of the genome is considered except for the first and last 75nt, which are left unaltered as they contain the conserved influenza promoter and regions considered to be critical for packaging. Therefore, for many lengths, we sample over multiple different permutations of junction identity, effectively correcting for how any given deletion sequence may influence replication independent of length. We reliably recover replication-competent templates ranging in size from 200-1500nt in HA, and 200-2700nt in PB1, which includes duplications in the latter that extend beyond the wild-type length.

Following our prior methods, we transfected these libraries with the minimal replication machinery, either with, or without, an NS2 expression vector. The frequency of variants of any given length in the population vRNA was measured under each condition by RNAseq, and its enrichment or depletion presented in Fig. 3. We see, in agreement with our previous results, a simple inverse relationship between segment length and replication in the absence of NS2—shorter molecular species replicate better than their longer counterparts (Fig. 3, top)[36]. Adding NS2, we see this relationship deteriorate for PB1 at the smallest lengths we test, but not for HA (Fig. 3, middle). Plotting the difference, we see that expression of NS2 dramatically reduces replication of PB1 variants ~200nt in length, and provides a slight, but consistent, selection for longer lengths in HA (Fig. 3, bottom). Both datasets closely match what we observed with individual variants by qPCR.

These data appear more compatible with the hypothesis that the expression of NS2 leads to sequence-, or site-specific, differences in replication rather than a length-dependent effect. Altering our analyzes, we now assess how the presence or absence of any given region in each segment impacts replication fitness, after correcting for length, presented in Fig. 4a. To understand our plots, each point represents the log fitness difference as a region is lost. If a region is required for efficient replication, values are negative, if it is somehow inhibiting replication, values are positive.

In the absence of NS2, we do not see significant contributions of any particular region of PB1 to replication efficiency. In contrast, regions adjacent to the 5' end of HA appear to negatively contribute to replication, although we do not investigate this phenomenon any further (Fig. 4a, top). As we add NS2, in HA we do not see any significant differences in replication efficiency associated with the loss of any particular genomic region. In stark contrast, we observe that regions in the 5' end of PB1 are absolutely required for efficient replication when NS2 is expressed (Fig. 4a, middle). Plotting the differences between replication with, and without, NS2, the NS2-specific contribution of the 5' end of PB1 to replication is quite apparent (Fig. 4a, bottom).

As sequence and length are difficult to untangle, we wished to explore the strong effect that we see around 100nt from the 5' of the PB1 segment further. To do so, we identify all deletions in our library that lack this region, and compare them to every sequence that contains this region, after matching each by length (Fig. 4b). If we are correct and this region is, generally, required for replication in the presence of NS2, and not due to some unusual structure created by a

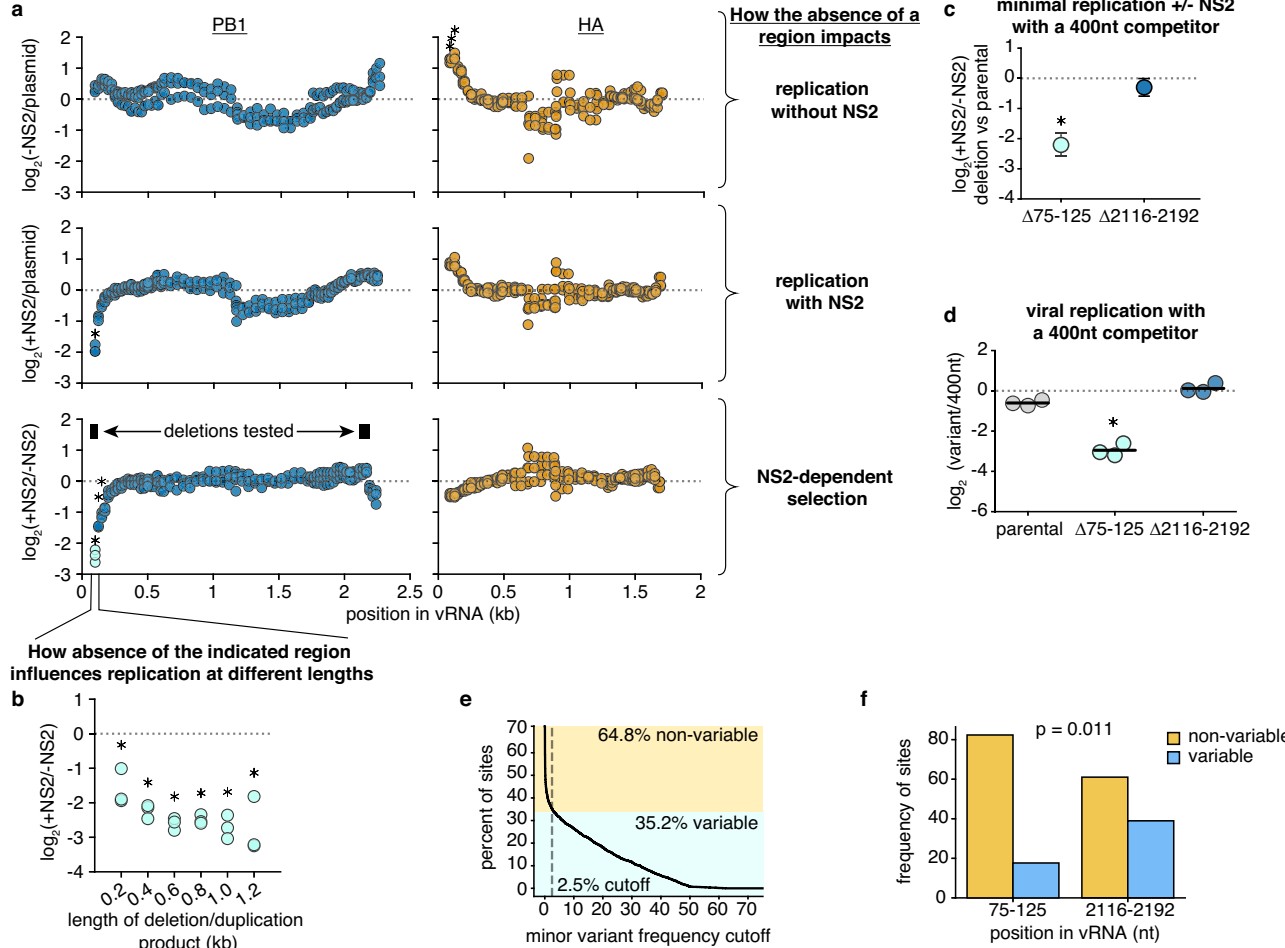

**Fig. 4 | NS2 expression suppresses replication of PB1 lacking certain genomic regions. a** Reanalysis of data from Fig. 3. Comparing variants of identical length, the frequency at which a region is, or is not, observed under each condition was determined, and the median across all lengths for a given replicate and region presented. The dotted line separates regions whose absence enhances replication (above) from those whose absence reduces replication (below). Asterisks indicate a significant effect that was greater than 2-fold, one-sample two-tailed t-test, Benjamini-Hochberg corrected FDR < 0.1. **b** How the absence of the indicated region in (**a**) influences replication upon NS2 expression across a range of different lengths. Points shown if there were at least 100 measurements across all conditions. Statistics performed as in (**a**), with a FDR cutoff of 0.05. **c** Each indicated deletion was generated in a PB1$_{177:385}$ background, co-transfected with a barcoded PB1 400nt competitor with, or without, NS2, and its replication was compared against the parental background at 24 h post-transfection by qPCR. Asterisks indicate that a given deletion reduces replication, one-tailed one-sample t-test, p < 0.05. Points

represent mean and standard deviation, n = 3. **d** Variants from (**c**) were co-transfected with a barcoded PB1 400nt competitor and rescued into virions using infectious virus. A549 cells were infected with this population at a genome-calibrated MOI of 25 for 8 hours. The change in competitor and variant frequencies in vRNA during viral replication was determined by qPCR. Asterisks indicate reduced replication of the indicated variant when compared to the parental, two-tailed two-sample t-test, Benjamini-Hochberg corrected FDR < 0.05. **e** Positions in a full-length PB1 alignment defined as variable or non-variable by calculating the fraction of nucleotides at a position that do not match the major variant. At the chosen cutoff, 2.5%, a position is considered variable if at least 2.5% of sequences do not match the major variant. This cutoff was chosen to match the inflection point on our curve. **f** Frequency of variable sites within the given regions of PB1 as defined in (**e**). Significant difference between categories tested using Fisher's exact test. Source data are provided as a Source Data file.

particular deletion, we would anticipate that no matter what length we explore, the absence of this region associates with decreased replication. As we would hope, when we explore our library data further, the absence of the region immediately proximal to the 5′ end of the PB1 vRNA was consistently associated with reduced replication when NS2 was expressed.

To further confirm this finding, we used a 562nt PB1 template, PB1$_{177:385}$, based on a DVG which we have previously characterized from a natural diversity, high MOI serially passaged viral population[36,48]. As it derives from a naturally-occurring DVG, we know it replicates robustly and is packaged effectively. We chose to use this DVG as it replicates more consistently than full-length PB1 in minimal replication assays (Supplementary Fig. 5). This segment acted as a parental vRNA from which we generated further deletions in an apparent NS2-dependent (75-125nt), and NS2-independent

(2116-2192nt) region. After generating these variants, we performed minimal replication assays with PB1$_{177:385}$, or each individual deletion, separately. In keeping with our goals of analyzing replication kinetics, we added an additional, barcoded, 400nt competitor to each of these experiments. Comparing against parental PB1$_{177:385}$, we find that a 5′ deletion does, specifically, impact replication when NS2 is expressed (Fig. 4c). Moreover, when we test these same variants in an infection model against the same 400nt competitor, we find that during true viral replication this deletion is also associated with a considerable drop in replication efficiency (Fig. 4d). As we might expect, when we explore naturally occurring variation we find that there is significantly less diversity observed in the region required for efficient replication in the presence of NS2 as compared to the region where we see little selection during replication (Fig. 4e, f).

Finally, we wished to understand whether we might be able to explain why replication of the HA segment exhibits a slight preference for longer variants when NS2 is expressed, but replication of PB1 does not. We hypothesized that the strong effect of the 5′ sequence in PB1 on NS2-dependent replication might mask the much more subtle effect we observe in HA. To test this hypothesis, we excluded any variant that removed the first 400nt, or last 175nt, from PB1 or HA, and repeated the analysis from Fig. 3 to measure NS2-dependent selection. We find no evidence to support our hypothesis, as while we still see that expression of NS2 preferentially supports the replication of longer variants of HA, no such effect is observed in PB1, indicating that simple increases in processivity is unlikely to be the main driver of our observations (Supplementary Fig. 6).

## Sites beyond the promoter modulate NS2-dependent replication

From these data it appears that NS2 expression affects replication efficiency in response to particular genomic regions or specific key sequences, rather than length. Therefore, we wanted to explore these interactions at the nucleotide-level in a single influenza segment. Also, as the first and last 75nt of the 5′ and 3′ termini were excluded in our prior analysis, we sought to extend our understanding to include these regions. These regions contain the U12 and U13 canonical influenza promoter sequences that are essential for genome (vRNA) and anti-genome (cRNA) production, as well as adjacent noncoding and coding sequences that are critical for the successful packaging of vRNPs into the virion.

To accomplish this, we performed random mutagenesis to introduce mutations into PB1$_{177:385}$. We chose this template for two reasons. First, central positions do not appear to strongly influence replication in the presence or absence of NS2 in PB1 (Fig. 4a). By removing these positions from our analysis, we reduce the number of sequences required to appropriately sample our library, improving accuracy. Second, as in our deletion analysis, using a template that is well-replicated provides us with a greater consistency in our measurements.

Our PB1$_{177:385}$ library was generated using error-prone PCR to introduce a per-site mutation rate of ~0.8%. (Supplementary Fig. 7) We opted for this rate rather than the goal of only a single mutation per template to achieve variation well-above the Illumina sequencing error rate (~0.1%). We assume that the majority of sites likely exhibit no effect, and so it is unlikely for epistasis to significantly complicate our measurements.

We then used this library in minimal replication assays with (+NS2), and without (−NS2), additional expression of NS2, and compared against our initial library (plasmid) or one another. We sequenced the RNA from these experiments using two different methods. First, we performed traditional amplicon sequencing by converting RNA to cDNA using a 3′-specific vRNA primer. Second, we performed 5′RACE (rapid amplification of cDNA ends) to analyze the first and last 20nt that are typically lost in the reverse transcription primer during amplicon-sequencing (Supplementary Fig. 8)[49]. A consideration when exploring our data is that measurements at the first 20nt are from vRNA templates, measurements at the last 20nt are from cRNA templates, and measurements at the internal sites represent predominantly vRNA sequence with some low level of contamination from cRNA and mRNA sequence. All data presented in this section refer only to genome or antigenome production, as we are focusing our study on genome replication and not transcription.

As before, we measured how each individual variant competes with the total library during genome replication with, or without, NS2. For each nucleotide position we generated two summary statistics, either measuring the collective enrichment or depletion of all non-wild-type nucleotides, or measuring the per-site information content (sites that prefer only a single nucleotide, for instance, have a high

information content). The two values relate to each other as follows: high information content sites allow for efficient replication only when these sites have a specific nucleotide present at that position, whereas our selection against non-wild-type sequences exhibits large negative values when replication specifically requires the wild-type nucleotide. Therefore, sites with high information content will likely exhibit large negative selection, provided of course that the wild-type nucleotide is preferred at that position.

Before exploring NS2-dependent effects on replication, we first wished to establish that all of our measurements were consistent and biologically relevant, as we are ultimately measuring the impacts of 2248 different possibilities when considering all positions in our template. Several features broadly give us confidence. First, our measurements were highly replicable across our independently-generated libraries (Supplementary Fig. 9 and Supplementary Fig. 10). Next, moving beyond technical repeatability, we wished to compare our measurements against prior knowledge. We would broadly anticipate the U12 and U13 promoter regions to be intolerant of substitutions, and that most other positions in the genome would not dramatically influence replication[23–25]. Comparing across all metrics, whether we consider information content or selection on non-wild-type nucleotides, we find that selection is considerably more stringent on promoter nucleotides than non-promoter nucleotides (Supplementary Fig. 11a,b). Moreover, when we explore our information content analysis to determine the identity of nucleotides in the promoter that support the highest levels of replication, we find that in the presence of NS2 the wild-type sequence is preferred at all sites (Supplementary Fig. 11c). Lastly, we do find that not all positions in the U12/U13 exhibit the same level of stringency. To confirm that there are positions in the promoter where substitutions do not completely abrogate replication, we introduced what we predict to be the most conservative mutation into one of the less stringent sites in the U13 promoter (G5U). In doing so, we find evidence that the resulting template can still produce replication-competent genomes in a minimal replication assay, confirming that some substitutions in the U13 do not completely remove promoter function and that we might expect to see differing levels of selection throughout the viral promoter (Supplementary Fig. 11d).

Having established that our assay appears to appropriately capture real biology, we next moved to assess how each position modulates replication in response to the expression of NS2 (Fig. 5a). As we explore the effects of NS2 expression on replication, we first note that the 75-125nt region we have already defined as impacting NS2-dependent replication kinetics has several sites (82, 83, and 91) which, when mutated, significantly impact replication in the presence of NS2—these, and adjacent positions, likely explain our results regarding deletions in this region (Figs. 5a, Supplementary Table 1). We further note that all 27 sites we identify as significantly modulating replication in the presence of NS2 are highly conserved across PB1 sequences from diverse IAV strains (Fig. 5b). However, despite their conservation between PB1 segments of diverse IAV strains, many of these sites are not absolutely conserved between IAV segments (Supplementary Fig. 13). This indicates that NS2-dependent effects on replication likely exhibit different requirements between different IAV segments.

Moving on from the 75-125nt region, and focusing on the strongest effects we observe, we find that these tend to be more restricted to either end of the viral genome. Mutations appear to negatively impact NS2-dependent replication at sites in the cRNA/mRNA promoter and a number of other positions adjacent to both the cRNA/mRNA promoter and the vRNA promoter. We present both the first, and last, 30nt of our template to better display these positions (Fig. 5c). We see similar results in our information content analysis, which considers selection across all four possible nucleotides at each position (Supplementary Fig. 12). Altogether, these findings indicate that, under polymerase activity supported by NS2, sequence identity

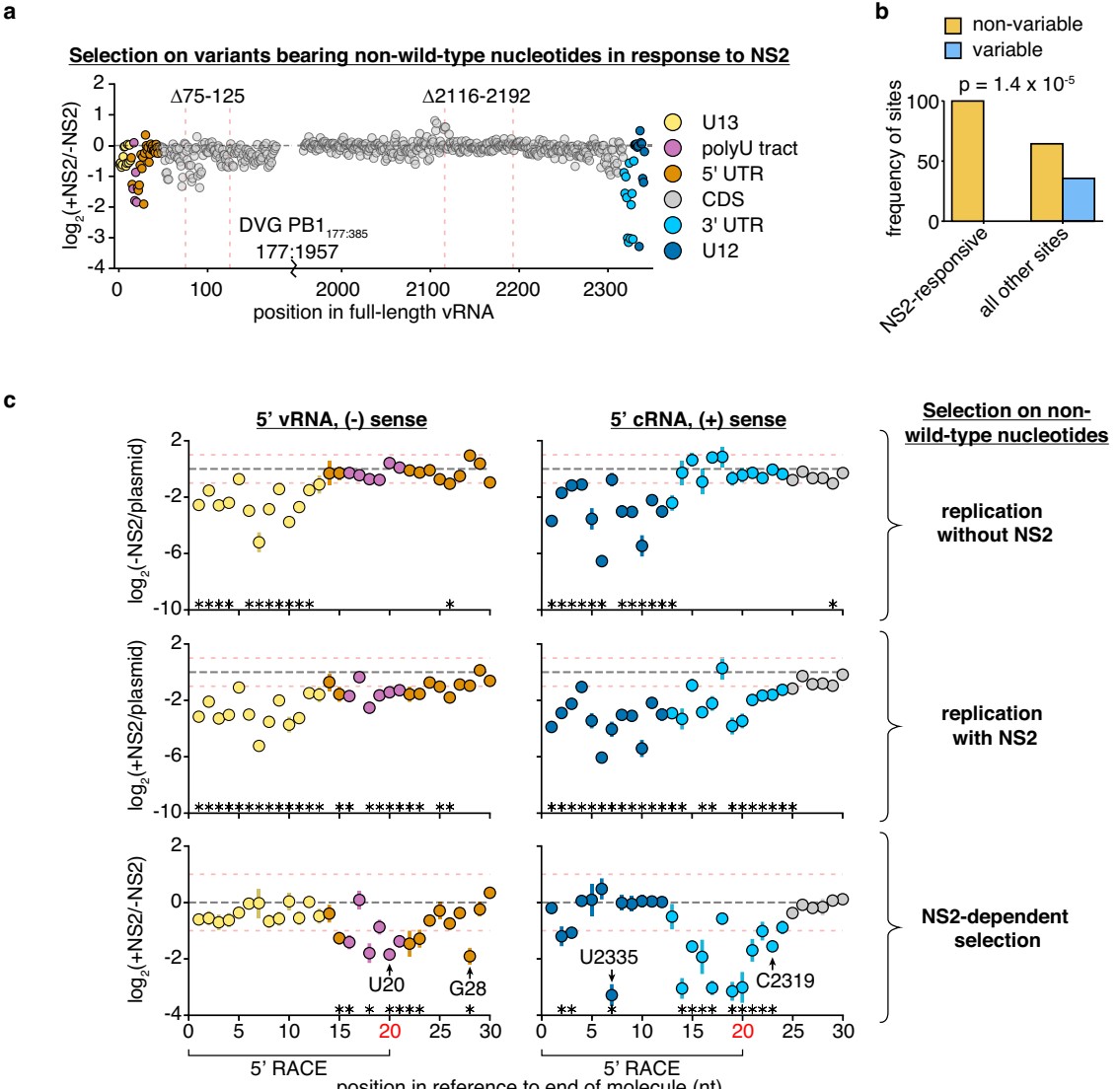

**Fig. 5 | Positions within and beyond the canonical promoter are required for efficient replication in the presence of NS2.** The frequency of total non-wild-type nucleotides at each position during minimal replication assays was measured under each condition, and its enrichment, or depletion shown. Similar data were procured for total information content, which considers selection at each individual nucleotide at each individual position (Supplementary Fig. 12) (**a**) NS2-dependent selection measured across all of PB1$_{177:385}$ against non-wild-type nucleotides, average value across all three replicates provided. Coordinates of deletions from Fig. 4 noted. Coordinates given as the full-length vRNA, regions of functional interest annotated. **b** Sites of significant selection in (**a**), as listed in Supplementary Table 1 analyzed as in Fig. 4f. Significant difference between categories tested using Fisher's exact test. **c** Selection on non-wild-type nucleotides in the first 30nt in the vRNA (left) and cRNA (right). Mean and standard deviation graphed. Asterisks indicate regions with a greater than 2-fold effect size (denoted by red dotted lines) that differ significantly from no effect, one-sample t-test, Benjamini-Hochberg corrected FDR < 0.1. The first 20nt of vRNA and cRNA were inferred from 5' RACE rather than simple amplicon sequencing (Supplementary Fig. 8). Positions further explored in Fig. 6 noted. Inter-replicate correlation plots presented in Supplementary Fig. 10. Source data are provided as a Source Data file.

beyond the canonical promoter is critical for efficient genome replication.

We next sought to validate our findings. We chose four positions where mutations significantly affected replication in response to NS2, two that were covered by RACE sequencing, and two that were covered by amplicon sequencing (Fig. 5c). Additionally, these four positions cover a range of different functional annotations, including a position within a tract of uracils critical for stuttering and generating poly-adenylated viral mRNAs[50–52]. Into each of these positions, we introduced a transversion mutation into PB1$_{177:385}$. We tested each mutation in minimal replication assays with, and without, NS2, co-transfected with a barcoded 400nt PB1 internal competitor (Fig. 6a). Our individual experiments matched our expectations from our library work, as each of these mutations in PB1$_{177:385}$ led to a reduction of replication fitness

in an NS2-dependent fashion when compared against their parental segment.

With these individual mutations in-hand, we wanted to know whether we have dramatically influenced the ability of NS2 to modulate vRNA/cRNA/mRNA ratios, or whether our mutations instead have more subtle effects on replication kinetics. We tested each individual variant in minimal replication assays with, and without NS2 and measured each molecular species (mRNA/cRNA/vRNA) by primer-extension (Fig. 6b). Consistent with prior reports, wild-type sequence showed a reduction in relative mRNA levels and an increase in cRNAs and vRNAs with NS2 expression[4].

As we extend our observations to our mutations, we see that all retain a reduction in mRNA levels upon NS2 expression (Fig. 6b, top). This difference is more dramatic for C2319G and U2335G, and U20A

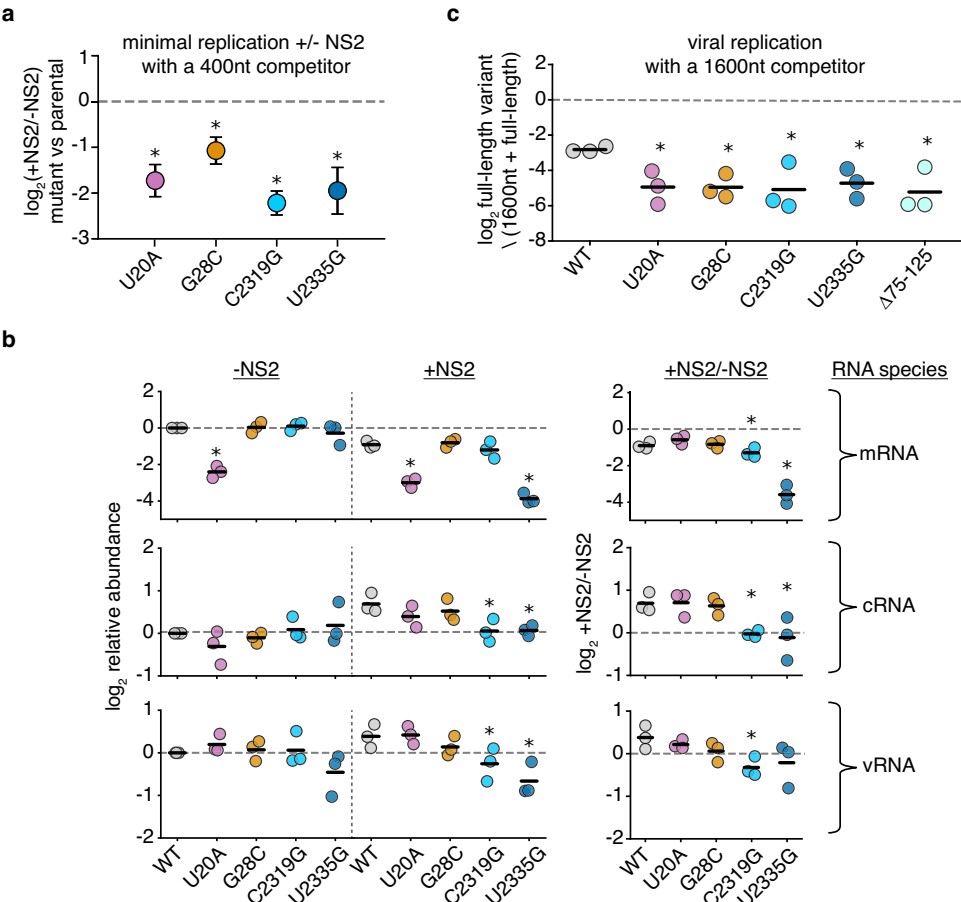

**Fig. 6 | Mutations to NS2-dependent sites can influence mRNA/cRNA/vRNA ratios and impact replication during infection. a** Each individual mutant in PB1$_{177:385}$ was transfected alongside a 400nt internal control and minimal replication machinery with, and without, NS2. The relative frequency of each variant relative to the 400nt control was compared against wild-type under each condition at 24 hours. Asterisks indicate a ratio significantly less than 1, one-tailed one-sample t-test, Benjamini-Hochberg corrected FDR < 0.05, indicating that the parental exhibits an NS2-dependent advantage over the variant. Points represent mean and standard deviation, $n = 3$. **b** Quantitative analysis of primer extension in PB1$_{177:385}$. All values in left two columns corrected against a parental template in the absence of NS2. Dotted line represents that value, points above indicate an increase in that molecular species, below, decrease. Values in the right column represent the ratio of points between the left two columns. Asterisks indicate values that are

significantly decreased relative to the parental template, one-tailed t-test with Benjamini-Hochberg corrected FDR < 0.1. Individual replicates and mean presented, $n = 3$. Similar analyzes for full-length PB1 template presented in Supplementary Fig.14. **c** Indicated variants were introduced into full-length PB1 bearing inactivating mutations to its start codon and rescued via reverse genetics alongside a 1600nt internal competitor. MDCK-SIAT1 cells expressing PB1 were infected with the resulting mixture at a genome-calibrated MOI of 25 for 14 hours. The frequency of each variant in the mixed population was measured before, and after, viral replication. Values plotted represent the change in frequency during replication only. Asterisks indicate values significantly less than the parental, one-tailed two-sample t-test with Benjamini-Hochberg error correction FDR < 0.05. Source data are provided as a Source Data file.

exhibits significantly reduced mRNA regardless of NS2 expression–disruption of the stuttering mechanism tied to the polyU tract could be resulting in aberrant, reduced, or absent polyA tailing and a subsequent decrease in mRNA stability[53,54]. For cRNA and vRNA, neither 5′ mutation (U20A or G28C) produces a significant change in absolute amount or NS2-driven dynamics, at least within the sensitivity of this assay (Fig. 6b, middle, bottom). This finding may indicate that the phenotypes we observe during competition-based assays are due to differences in the relative kinetics of vRNA or cRNA production upon NS2 expression, but do not completely remove the ability of NS2 to influence replication of these templates. For C2319G, and U2335G, however, we observe that there is no apparent increase in cRNA production upon NS2 expression, and a reduction in both cRNA and vRNA. Therefore, these two templates appear to break the ability of NS2 to modulate the production of cRNA, and, given our results on vRNA, indicate the presence of NS2 can actually suppress replication of some templates.

To determine if these effects were unique to the template we tested, we also performed these analyzes on full-length PB1 (Supplementary Fig. 14). While replicating this template, which may be under different kinetic constraints than our shorter template, we see similar, although not identical, results. When examining vRNA and cRNA, no mutation results in significantly reduced template in the absence of NS2 (although curiously U2335G enhances cRNA production). When NS2 is expressed U20A dramatically reduces vRNA and cRNA, G28C and C2319G reduce cRNA, and U20A, G28C, and U2335G do not exhibit as significant an enhancement of cRNA production. Notably, each mutation influences genome concentration negatively in some manner, although it is difficult to interpret these results, likely in part as it is unclear how the concentration of each product depends on kinetic rate of generation versus other, unconsidered, features, in our assays. To reiterate, the differences we observe between full-length and our shortened templates likely reflect differing requirements for the final, steady-state concentrations observed in these assays, but we see with

confidence that each and every mutation perturbs replication in some manner on at least one template tested.

Most importantly, we sought to answer the question of whether the NS2-responsive positions we identify in competitive minimal replication assays with PB1$_{177:385}$ are relevant to replication of the natural template during infection. We rescued each mutation in full-length PB1 with a start codon mutation alongside a 1600nt competitor, including our deletion from nucleotides 75-125. A key aspect of these rescues is that neither template provides PB1 protein, removing considerations of feedback loops or influence on mRNA production during infection. To introduce these mutations into wild-type influenza, without such perturbations, would unfortunately produce uninterpretable results owing to the impact on PB1 expression, which itself would influence genome replication. We also provide an internal competitor unable to express PB1, such that competition between our variants of interest and this segment does not itself also feed back on PB1 expression, leading to complex, uninterpretable, results. This protein is instead expressed ectopically by MDCK-SIAT1 cells, which we used as a host in these experiments. Measuring a single-round of viral replication by using a high MOI infection containing these viruses and sampling cellular vRNA, we observe that each mutation dramatically impacts the ability of this template to replicate. Thus, the sites we identify in our minimal replication assays, even those incredibly distal to the promoter, are relevant to genome replication during viral infection (Fig. 6c).

## Discussion

Our study redefines the minimal sequence requirements for efficient replication of the IAV genome. These requirements were masked in earlier studies as they lacked NS2, a key factor promoting the replicative form of the IAV polymerase. This explanation is in accordance with two other reports that found positions that modulate replication outside of the core promoter, but did so either in the context of infection, where NS2 would be present, or in the NS segment, where NS2 would also be expressed[55,56]. Additionally, the competitive nature of our assays, even in our reconstruction of individual variants of interest, allows us to capture kinetic differences that may not be apparent in steady-state, endpoint, assays (Supplementary Fig. 1). External to the canonical U12 and U13 promoters, we have now found that sequence identity at positions in the PB1 segment in the polyU stretch, 5' and 3' UTR, and even some regions within the coding sequence itself, are all required for efficient viral replication in the presence of the polymerase accessory factor NS2. Even within the U12 promoter, we find a single position where variants exhibit extremely different behavior during replication with, and without, NS2. These data, collectively, not only explain how NS2 can influence the spectrum of defective species observed during viral replication due to differential selection during genome replication, but also, more importantly, force us to reconsider selection across the genome during replication.

The role of NS2 in genome replication appears to be to promote the replicative, and suppress the transcriptional, form of the polymerase. A key difference in the two forms is the latter associates with the C-terminal domain of host RNA polymerase II to engage in cap-snatching of nascent transcript and acts as a single heterotrimer, whereas the former instead associates with host factors such as ANP32A and acts as a larger order oligimer (canonically thought to be a dimer)[12,14,57–61]. Explaining in large part how NS2 accomplishes this feat, a recent cryo-EM structure of an NS2-polymerase hexamer, consisting of three dimers of NS2 and three dimers of the heterotrimeric IAV polymerase, reveals that NS2 promotes multimerization, and, while doing so, sterically blocks the site at which the IAV polymerase engages with host RNA polymerase II[8]. Consistent with the physical block of transcription by NS2, while we find several sites that appear to remove the ability of NS2 to support cRNA production, none remove the ability

of NS2 to decrease transcription. These are not the first mutations that can uncouple transcription and replication, for instance removal of an unpaired adenosine in the 5' end of the vRNA within the U12 sequence decreases transcription while leaving replication intact, although they are the first that have been linked to this phenomenon of enhanced replication upon NS2 expression[62].

Supporting our experimental work, we observe that the sites we identify as strongly influencing replication in the presence of NS2 are broadly conserved in the PB1 segment. For many of these sites, this conservation was previously noted, but attributed to packaging requirements[33,63–65]. From our work, we could draw two different conclusions: 1) the first is that, as is likely true with our mutation in the polyU tract, these sites are conserved for multiple, potentially independent, reasons, and so may be highly constrained, or 2) early measurements defining these regions as involved in packaging were often based on expression or incorporation within virions with an underlying assumption from minimal replication assays that they did not concurrently influence genome replication. As we now know that this assumption may not be correct, it may be worth re-examining some prior assumptions regarding packaging that may be better explained by an inability to efficiently replicate. We note our data by no means preclude these sites from engaging in packaging interactions, but rather provide a caution that other pressures may have produced an apparent inability to package.

Lastly, we would like to contextualize how and why NS2 might possibly engage with the sequences we identify. It is not thought that NS2 can bind to RNA directly[2]. In its role as an export protein, it is thought to interact with M1, which interacts with nucleoprotein, which in turn interacts with the viral RNA[10,66]. In its role in promoting replication, it has been shown to directly bind to the heterotrimeric IAV polymerase, as discussed above[8]. What, then, explains our findings? The most parsimonious explanation which we can provide is that what we are measuring is sequence constraints on replication by the IAV polymerase in a more "true" replicative conformation. Therefore, it is not so much that NS2 directly interacts with the sites we describe, but rather the conformation the IAV polymerase adopts upon NS2 expression is more sensitive to sequence identity at these individual sites. With the recent structure of the IAV polymerase in complex with NS2, we are optimistic that our results, along with the hard work from other groups, may provide a more complete, and nuanced, molecular model of IAV replication and its sequence requirements. This is of particular importance given the role of this step in defining host range, as both host cofactors and viral, including NS2, influence the ability of IAV to transition from avian to human hosts[20]. Collectively, we hope that our data will help us all to better understand the nature and evolutionary trajectories of this important human pathogen.

## Methods

### Cell lines and viruses

The following cell lines were used in this study: HEK293T (ATCC CRL-3216), MDCK-SIAT1 (variant of the Madin Darby canine kidney cell line overexpressing SIAT1 (Sigma-Aldrich 05071502)), MDCK-SIAT1 PB1, and A549 (human lung epithelial carcinoma cell line, ATCC CCL-185)[48]. Cell lines were tested for mycoplasma using the LookOut Mycoplasma PCR Detection Kit (Sigma-Aldrich) using JumpStart Taq DNA Polymerase (Sigma-Aldrich). All cell lines were maintained in D10 media (DMEM supplemented with 10% heat-inactivated fetal bovine serum and 2mM L-Glutamine) in a 37 °C incubator with 5% CO2. Save where specified, all IAV experiments used A/WSN/1933, or sequences derived from the same. Genome sequence of A/WSN/1933 provided in Supplementary Data 1. Sequence of PB1$_{177:385}$ provided in Supplementary Data 2. Sequences of NS2 from other IAV strains provided in Supplementary Data 3.

Wild-type A/WSN/1933 (H1N1) virus was created by reverse genetics using plasmids pHW181-PB2, pHW182-PB1, pHW183-PA,

pHW184-HA, pHW185-NP, pHW186-NA, pHW187-M, and pHW188-NS[67]. HEK293T and MDCK-SIAT1 cells were seeded in an 8:1 co-culture and transfected using BioT (Bioland Scientific, LLC) 24 hours later with equimolar reverse genetics plasmids. 24 hours post transfection, D10 media was changed to Influenza Growth Medium (IGM, Opti-MEM supplemented with 0.04% bovine serum albumin fraction V, 100 μg/ml of CaCl2, and 0.01% heat-inactivated fetal bovine serum). 48 hours post-transfection, viral supernatant was collected, centrifuged at 300 g for 4 minutes to remove cellular debris, and aliquoted into cryovials to be stored at −80 °C. Thawed aliquots were titered by TCID50 on MDCK-SIAT1 cells and calculated using the Reed and Muench formula[68]. To create wild-type viral stocks with a low defective content, MDCK-SIAT1 cells were infected at an MOI of 0.01 and harvested 30 hours post-infection and titered. For mixed viral stocks in the experiment described in Fig. 6c, the indicated variants were used instead of pHW181-PB1, an expression construct for PB1 under control of a CMV promoter was additionally transfected, and previously-described MDCK-SIAT1-PB1 cells were used instead[48]. These viral stocks were harvested after 72 hours rather than 48.

### Reverse transcription and qPCR
For all qPCR experiments, cDNA was generated using the High Capacity First Strand Synthesis Kit (Applied Biosystems, 4368814). Reverse transcription was performed using the manufacturer's protocol, save for instead of the provided primers, we used the universal vRNA primers as described in Hoffmann et al.[69]. An exception is the *L32* measurements, where we used random-primers included with the kit instead. Reaction conditions were an initial step of 10 minutes at 25 °C, reverse transcription at 37 °C for two hours, and an 85 °C heat inactivation step for five minutes. All qPCR experiments used Luna Universal qPCR Master Mix (NEB #M3003), with primers specified in Supplementary Data 4. qPCR conditions for all experiments were as follows; denaturation at 95 °C for one minute, then cycles of denaturation at 95 °C for 15 seconds, and annealing and extension at 60 °C for 30 seconds. A melting curve was then performed to ensure all products are mono-peaked. Controls for each qPCR varied, for length-variant measurements the barcode associated with 1600nt variants was used as our control, for all other measurements we used a qPCR specific to a barcode associated with a 400nt variant in PB1. RNA was purified for all experiments either using the RNeasy plus mini kit (Qiagen, 74134), or the Monarch Total RNA Miniprep Kit from New England Biolabs (NEB #T2010). For measurements of cell-free viral populations, 100 μl of supernatant and 600 μl of lysis buffer was combined and processed. All primer pairs for all qPCR presented in Supplementary Data 4.

### Mixed variant population virus and single cycle infection
Barcoded length variant segments of 200, 400, 800, and 1600nt in length in PB1 and HA were described previously[36]. To generate viral populations of PB1 and HA containing the above-mentioned length variants or, for PB1, the variants described in Fig. 4d, we first transfected an 8:1 co-culture of 400,000 HEK293T and 50,000 MDCK-SIAT1 cells in a 6-well plate with mixed variant populations for either PB1 and HA at equimolar ratios. In this same transfection, we included pHDM plasmids encoding mRNA for PB2, PB1, PA, and NP downstream of a CMV promoter, also in equimolar ratios. BioT was used as our transfection reagent as per the manufacturer's protocol. After 24 hours, D10 media was replaced with IGM and infected the cells with a low-defective wild-type WSN population at an MOI of 0.25 for 72 hours. The virus-containing supernatant was purified by centrifugation at 300 g for 4 minutes. Initial viral population composition was measured by qPCR.

A549 cells were then infected with mixed populations consisting of 200nt, 400nt, 800nt, and 1600nt variants of either HA or PB1 with complementing wild-type virus at an MOI of 25, as measured by

genome equivalents by qPCR. Cells were then washed to remove any additional virus at 2 hours post-infection, and, at 8 hours post-infection supernatant was removed, cellular RNA harvested, and qPCR used to measure the relative frequency of each variant. Each measurement was then corrected by the 1600nt barcode or 400nt barcode, as indicated in figure captions.

For experiments in Fig. 6c, a similar protocol was followed, however wild-type WSN was not used but instead a mixed population of each indicated variant and a 1600nt barcoded control was rescued using MDCK-SIAT1 cells expressing PB1 and the inclusion of a PB1-expression construct in the initial transfection. Identically, qPCR was used to measure the ratio within an initial population prior to infection, and numbers presented represent the change in this ratio after 14 hours at an infection initiated at an MOI of 25 in MDCK-SIAT1 cells expressing PB1.

### Minimal replication experiments for individual variants
For all experiments, 100,000 HEK293T cells were seeded in a 24-well plate the day prior. Indicated vectors were co-transfected using BioT into our cells with equimolar amounts of plasmids encoding the minimal replication machinery (PB1, PB2, PA, and NP), with, or without, the addition of NS2. Expression constructs for minimal replication machinery for these experiments were in a pHDM backbone downstream of a CMV promoter and a β-globin intron. 24 hours post transfection, cells were harvested, RNA purified, and qPCR used to assess the frequency of each variant after replication.

For length-variant experiments in Figs. 2 and Supplementary Fig 2 200, 400, 800, and 1600nt variants of each A/WSN/1933 genomic segment were cloned into pHH21 for expression of authentic genomic RNA[70]. Each of these variants possessed equal amounts of the 5′ and 3′ termini, and to each a previously-validated barcode was cloned such that they could be disambiguated by qPCR, with equivalent qPCR primers even between segments[36]. Each variant also had the canonical start codon mutated to GTA. All clonings (including removal of start codons) used an inverse-PCR strategy and subsequent re-closing with Gibson. Primers for these clonings are provided in Supplementary Data 4. For experiments in Supplementary Fig. 2, NS2 was cloned from each indicated strain into a pHDM expression vector. Sequences for NS2 variants provided in Supplementary Data 3.

For experiments in Figs. 4c and 6a, each variant was constructed by inverse PCR using primers described in Supplementary Data 4. Transfection of each variant was performed alongside co-transfection of a barcoded 400nt variant of PB1, as described in the prior paragraph. Ratio of each indicated variant to this control was measured under each condition, and compared against the same ratio from an experiment with the parental template. Error for these experiments was calculated using standard propagation of error.

Lastly for the experiment described in Supplementary Figs. 5 and Supplementary Fig. 11d experiments were performed as above, save no internal control was included and experiments were performed with, and without, the inclusion of the PB1 protein expression construct. Specific primers were used for Supplementary Fig. 11d that cannot amplify any product from the PB1 protein expression construct to ensure that the activity we measured was specific to activity on our template, and are described in Supplementary Data 4.

### Mutagenesis library construction
In triplicate, a purified plasmid encoding PB1$_{177:385}$ was used as a template with the GeneMorph II Random Mutagenesis kit (Agilent, #200550) in an error-prone PCR, with primers 5′-GGTCGACCTCCG AAGTTGGG-3′ and 5′-TTTTGGGCCGCCGGGTTATT-3′. Following manufacturer's instructions to achieve a 1% error rate across the 562nt vRNA, we included 80 ng of target DNA (501.5 ng total plasmid DNA) in the mutagenic PCR reaction. Reaction conditions were 30 PCR cycles, a 54 °C annealing temperature, and 1 minute extension time. The

resulting product was run on an agarose gel and purified (New England Biolab's Monarch DNA Gel Extraction Kit, #T1020). 10 ng was subjected to an additional 10 cycles of amplification with Q5 Hot Start High-Fidelity 2X Master Mix (New England Biolabs #M0494) using the same primers as the mutagenic PCR, a 55 °C annealing temperature, and a 30 second extension time. The backbone for the mutagenic library was amplified with primers

5′-CTTCGGAGGTCGACCAGTACTCCGGTTAACTGCTAGCG-3′ and
5′-CCCGGCGGCCCAAAATGCCGACTCGGAGCGAAGGAGCGAAAG ATATAC3′

using Q5 Hot Start High-Fidelity 2X Master Mix and reaction conditions with a 55 °C annealing temperature, a 1 minute 30 second extension time, and 28 cycle repeats. Both the mutagenic PCR and backbone were run on an agarose gel, column purified, and bead cleaned with 3X beads:sample by volume (Promega ProNex Size-Selective Purification System, #NG2001). A 20 µL Gibson Assembly reaction was performed with 50 ng of vector at a 2:1 ratio for 1 hour (New England Biolabs NEBuilder HiFi DNA Assembly Master Mix, #E2621). Assembly reactions were bead cleaned with 1X beads:sample by volume to remove excess salts prior to electroporation in Electro-Max Stbl4 cells according to the manufacturer's protocol (Invitrogen, #11635018). Cells were spread across 15 cm LB Agar + 100 µg/ml Ampicillin plates and allowed to grow overnight at 37 °C. Libraries were scraped and resuspended in a 10 mL LB broth. Aliquots were centrifuged until bacterial pellets appeared to be of comparable size, then plasmids were purified from these stocks (New England Biolabs Monarch Plasmid Miniprep Kit, #T1010).

## Minimal replication experiments for libraries

For minimal replication assays involving the length polymorphic and mutagenesis libraries, 400,000 HEK293T cells were seeded in D10 in each well of a 6-well plate 24 hours prior to transfection. Cells were transfected with helper plasmids encoding only the mRNA for A/WSN/33 PB2, PB1, PA, and NP with or without the addition of NS2 (in pHAGE vectors downstream of a CMV promoter for libraries as in Mendes and Russell[36]), as well as the plasmid library in 1:1 ratios using BioT transfection reagent, according to the manufacturer's protocol (Bioland Scientific LLC, # B0101)[36]. 24 hours post-transfection, cells were harvested with 300 µL of RNA lysis buffer and purified using the Monarch Total RNA Miniprep Kit (New England Biolabs, #T2010).

## Sequencing preparation of mutagenesis libraries

Sequencing and analysis of length-variant libraries performed as described in Mendes and Russell[36]. For single-nucleotide variant libraries in PB1$_{177:385}$, for non-RACE sequencing we first converted RNA from our experiments to cDNA with a universal vRNA primer with adapter from Hoffmann et al., 5′-TATTGGTCTCAGGGAGCGAAAG-CAGG-3′ using Superscript III according to the manufacturer's protocol (Invitrogen, #18080400)[69]. PCRs were performed to amplify and append partial Illumina i5 and i7 adapter sequences across four subamplifications of each library to sequence the entire vRNA genome, except the terminal ends which cannot be captured with this method (see 5′ RACE sequencing methods). For all reactions, 1 µL of cDNA or 10 ng of plasmid were used as a template for each mutagenesis library with Q5 Hot Start High-Fidelity 2X Master Mix. The first PCR used universal primers for the PB1 segment from Hoffmann et al (5′-Bm-PB1-1-G4: 5′-TATTGGTCTCAGGGAGCGAAAGCAGGCA-3′; 3′-Bm-PB1-2341R: 5′-ATATGGTCTCGTATTAGTAGAAACAAGGCATTT-3′) with a 55 °C annealing temperature and 20 second extension time for 13 cycles, and was subsequently run on an agarose gel, extracted, and column purified. The second PCR appended Illumina intermediate sequences to staggered sub-amplifications (sequences provided in Supplementary Data 4) of each library with a 60 °C annealing temperature and 20 second extension time for 7 cycles. Sub-amplifications were pooled and purified with 3X beads:sample by volume.

To identify the sequences of the terminal ends of the vRNA and cRNA promoter that are lost during typical RNAseq amplification methods, the mutagenesis library samples were reverse transcribed with the Template Switching RT Enzyme Mix (New England Biolabs, #M0466) according to the manufacturer's protocol and subsequently amplified and appended with the partial Illumina i5 and i7 adapter sequences using Q5 Hot Start High-Fidelity 2X Master Mix. In brief, 4 µL of sample were annealed with the universal Hoffmann et al. primers for PB1 (5′-Bm-PB1-1-G4: 5′-TATTGGTCTCAGGGAGCGAAAGCAGGCA-3′ for U12 priming; 5′-TATTGGTCTCAGGGAGTAGAAACAAGGC-3′ U13 priming) and then reverse transcribed with the template switching oligo (TSO, 5′-Biotin-AAGCAGTGGTATCAACGCAGAGTACATrNrG+G-3′, containing mixed ribonucleotides at the 3rd from final position, a locked nucleic acid at the final position, and a 5′ biotin modification to prevent spurious additional concatemerization) similar to that from Picelli et al[71,72]. Amplification of the 5′ terminal ends of the vRNA and cRNA strands were performed in separate reactions using a TSO-specific amplification primer (5′-AAGCAGTGGTATCAACGCAGAGT-3′) and either a vRNA amplification primer (5′-CACCATGGATACTGTCAACAGG-3′) or cRNA amplification primer (5′-TCTGAGCTCTTCAATGGTGG-3′). Amplicons were run on an agarose gel, extracted, and column purified prior to appending partial Illumina i5 and i7 adapters (common TSO partial adapter:

5′-GTCTCGTGGGCTCGGAGATGTGTATAAGAGACAG AAGCAGTG GTATCAACGCAGAGT-3′; vRNA partial adapter: 5′-TCGTCGGCAGCG TCAGATGTGTATAAGAGACAGTCCAGAGCCCGAATTGATGC-3′; cRNA partial adapter:

5′-TCGTCGGCAGCGTCAGATGTGTATAAGAGACAGCCTGTTGAC AGTATCCATGGTG-3′). PCRs were run with a 65 °C annealing temperature and 20 second extension time for 15 cycles and were subsequently run on an agarose gel, extracted, and column purified.

To append final Illumina indices, a final PCR using Q5 Hot Start High-Fidelity 2X Master Mix was performed with a 62 °C annealing temperature and 20 second extension time for 7 cycles. Indexing primers were from IDT for Illumina DNA/RNA UD indexes, Set A, # 20026121. To ensure uniform amplification, samples were run on an agarose gel, extracted, column purified, and bead purified with 3X beads:sample by volume. Finally, sample purity and concentration were assessed via nanodrop and Qubit, respectively.

## Computational analysis

Assignment of barcodes in length-variant libraries was performed as described previously[36]. For single nucleotide variant libraries, amplicon and 5′ RACE analyzes proceeded with slightly different parameters. First, for amplicon analysis reads were first trimmed using Trimmomatic with the following parameters: 2 seed mismatches, a palindrome clip threshold of 30, a simple clip threshold of 10, a minimum adapter length of 2, keep both reads, a lead of 20, a sliding window from 4 to 15, and a minimum retained length of 36[73]. Reads were then mapped against PB1$_{177:385}$ using STAR, with relaxed mismatch parameters (permitting a mismatch frequency of 10% over the length of the read), requiring end-to-end mapping, and an intron of 1 to remove splice-aware alignment[74]. Nucleotide identity per site was then assessed using a custom script and a quality score cutoff of 30 (that is, sites below this score were not considered). Indels, while they are generated by the mutagenic PCR, were excluded from our analysis.

For 5′ RACE data, a number of reads were found to possess 5′ heterogeneous caps, likely either mRNA or capped cRNA sequences. These needed to be excluded such that identical molecular processes were being studied. For RNA sequencing, we enforced a match to either the sequence

5′-AAGCAGTGGTATCAACGCAGAGTACATGGGAGCGAAAGCAG GCAAACCAT-3′ or

5′-AAGCAGTGGTATCAACGCAGAGTACATGGGAGTAGAAACAA
GGCATTTTT-3′

containing no more than 5 differences, as these sequences span the TSO and 5′ end of vRNA/cRNA. To produce a comparable plasmid dataset we used the sequences

5′AAGCAGTGGTATCAACGCAGAGTCGGAGTACTGGTCGACCTCC
GAAGTTGGGGGGGGAGCGAAAGCAGGCAAACCAT-3′, and 5′ AAGCAGT
GGTATCAACGCAGAGTCTCCGAGTCGGCATTTT GGGCCGCCGGGTTA
TTAGTAGAAACAAGGCATTTTT-3′, to include the plasmid backbone and permitted only 4 differences as there is a variable nucleotide in the TSO that is unaccounted for in the plasmid data.

After filtering our reads in this manner, we used STAR to map against a custom reference consisting of PB1$_{177:385}$ with TSO adapters. Individual positions were then assessed as with our amplicon data. The first 20nt of our dataset were derived from 5′ RACE, remaining positions derived from amplicon sequencing.

For assessments of entropy and generation of sequence motifs for our mutational library, we used a calculation based on Shannon entropy with information content in bits as described by Schneider et al.[75,76]. This calculation consists of taking the total possible information content for a nucleotide position (2 bits) and subtracting the Shannon entropy. The latter is calculated by taking the sum of frequency that each nucleotide was observed multiplied by the $\log_2$ of that same frequency. However, instead of the frequency at which a base is observed, we scaled a presumed frequency of a nucleotide by the observed selection (that is we summed all selections, and took the relative ratio between any given selection and this summed selection). For instance, if there is no strong selection, and all nucleotides are observed at equal frequency before, and after, a given experiment, then each nucleotide is given a frequency of 25%. This would maximize Shannon entropy (2 - (0.25x−4)x 4, or −2), giving an information content of 0. Conversely, if other nucleotides were completely purged during selection, this would give us maximal information content of 2. Letter heights in sequence logos were procured by multiplying the contribution of a given selection-derived frequency of a given nucleotide by the total information content at a position. Sequence logo graphs generated by the package dmslogo (https://github.com/jbloomlab/dmslogo).

Lastly, for natural sequence diversity, we procured PB1 sequences from the NCBI flu database https://www.ncbi.nlm.nih.gov/genomes/FLU/Database/nph-select.cgi?go=genomeset on July 25th, 2024 that met the following prerequisites: full-length, 2341nt long, and exclude vaccine and laboratory strains. These sequences were then clustered by CD-HIT to an identity of 99%, and a single example of each cluster retained[77]. Thereafter, sequences with ambiguities or indels relative to other PB1 sequences, were manually removed. This resulted in 1939, high-quality, PB1 sequences that exhibit good alignment when simply lined-up with one-another, giving high confidence in the comparability of any given position, and, given the CD-HIT clustering, are unlikely to be aggressively weighted towards any single outbreak (Supplementary Data 5) At this point, no further alignment was required, and analysis proceeded.

All code was run on the San Diego supercomputer center, Triton shared computing cluster[78].

The following software and package versions were used: Python 3.12.2, Trimmomatic 0.39, STAR 2.7.11b, Samtools 1.19, Samtools 1.19, FastQC 0.12.1, numpy 1.26.4, matplotlib 3.8.3, seaborn 0.13.2, pandas 2.2.1, scipy 1.12.0, and statsmodels 0.14.1.

All computational analysis presented in https://github.com/Russell-laboratory/NS2_sequence_specificity, found at the doi generated by Zenodo, https://doi.org/10.5281/zenodo.14796919.

### Primer extension analysis

For all experiments in Fig. 6b and Supplementary Fig. 14, 100,000 HEK293T cells were seeded in a 24-well plate the day prior. Indicated PB1 variants in a pHH21 vector were co-transfected using BioT into our cells with equimolar amounts of plasmids encoding the minimal replication machinery (PB1, PB2, PA, and NP), with, or without, the addition of NS2. Expression constructs for minimal replication machinery for these experiments were in a pHDM backbone downstream of a CMV promoter and a β-globin intron. 24 hours post transfection, cells were harvested, RNA purified, and qPCR used to assess the frequency of each variant after replication. Unlike in qPCR experiments, only the indicated PB1 variant was transfected, with no additional internal control.

For RNA extraction, total RNA was extracted from cells lysed in Trizol using chloroform extraction and isopropanol precipitation as described previously[79]. ~400 μg of RNA was subsequently reversed transcribed using 32P-labeled PB1 c/mRNA-specific primer (PB1_146: 5′-TCCATGGTGTATCCTGTTCC-3′), 32P-labeled PB1 vRNA-specific primer (PB1_2203: 5′-CGAATTGATTTCGAATCTGG-3′), 32P-labeled 5S rRNA-specific primer (5S100: 5′-TCCCAGGCGGTC TCCCATCC -3′), and a SuperScript III reaction mix (Invitrogen #18080044) containing 1 U/μl RNase inhibitor (ApexBio #K1046). cRNA products were resolved using 7 M Urea/12% PAGE in 1x TBE.

Radioactive signals were collected using BAS-MS phosphorimaging plates (FujiFilm #28956475) and visualized using a Typhoon scanner. Densitometry analysis was performed using ImageJ.

### Reporting summary

Further information on research design is available in the Nature Portfolio Reporting Summary linked to this article.

## Data availability

The processed files and raw sequencing data generated in this study have been deposited in the NCBI gene expression omnibus under accession GSE276697, or additionally under bioproject PRJNA1158716. The qPCR data concerning confirmation of individual variants, whether by transfection or viral infection, primer extension analysis data, and analysis of sequencing data as presented in this manuscript are all provided in the Source Data file. Source data are provided with this paper.

## Code availability

The analysis code used in this study is provided at https://github.com/Russell-laboratory/NS2_sequence_specificity, archived at Zenodo under https://doi.org/10.5281/zenodo.14796919.

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

## Acknowledgements

This work was in part funded by the NIGMS of the NIH under grant R35GM147031 awarded to ABR and by NIH grant DP2 AI175474 awarded to AJWtV. KAR was supported by NIH training grant 1T32GM148739-01A1 and an NSF GRFP under grant number DGE–2039656. The funders had no role in study design, data collection and analysis, decision to publish, or preparation of this manuscript. We thank Rommie Amaro and Alma Castaneda for insightful discussions regarding IAV polymerase structure. We lastly thank our Summer SURF student Jeffrey Yumbla for his work on this project.

## Author contributions

A.B.R., M.M., S.S. designed the study, performed experiments, analyzed and interpreted the data, and prepared the manuscript and revisions. A.J.W.t.V. aided in study design, interpretation of data, manuscript presentation, and performed experiments. K.A.R., Y.Z., I.M. and M.G. performed key experiments. All authors reviewed this manuscript and provided useful feedback regarding overall presentation.

## Competing interests

The authors declare no competing interests.
