## [Transparent Peer Review file · Nature Communications]

Analysis of NS2-dependent effects on influenza PB1 segment extends replication requirements beyond the canonical promoter

Corresponding Author: Professor Alistair Russell

Version 0:

Reviewer comments:

Reviewer #1

(Remarks to the Author)

In this study, Swaminath and colleagues investigate the impact of the Influenza A virus NEP (formerly known as NS2) on the replication of individual viral genome sequences. They can show for a PB1 fragment that the impact of NEP on its replication depends on a sequence located at the 5' end, which is not part of the core promoter.

It has been known for several years that NEP, in addition to its role in the nuclear export of newly synthesized viral genome segments, is also a crucial co-factor of the viral polymerase. The underlying mechanism is still largely unknown. To investigate the influence of NEP on the replication of individual genome segments, the authors employ polymerase reconstitution assays. They use a short variant of the PB1 segment to identify critical sequence positions for NEP-dependent replication. Verification of these positions in a full-length PB1 segment was insufficiently performed. Furthermore, this reviewer feels that the relevance of the here presented results is limited without an evaluation of the critical positions in the context of an authentic viral infection.

The title of the manuscript is misleading. With the exception of a PB2 variant of 200nt, the presence of NEP results in an overall reduction of replication (Comp. Fig. 2A). Yet the authors claim in their title that efficient genome replication in IAV requires NEP. Further, the authors focus on an individual segment: PB1. Sequence-specific effects were not studied for other segments. This should be reflected in the title.

The authors use a 562nt PB1 fragment (PB1_177:385) in order to identify a critical sequence at the 5' end. Although the authors state that the PB1_177:385 fragment was chosen to increase the dynamic range of their measurements, it is yet very artificial. The authors should repeat the experiment shown in Fig. 4c in the context of full length PB1. Along these lines, the experiment shown in Figure 6a should be repeated in the context of full length PB1. Further a quantification of the primer extension analysis should not only be conducted for cRNA, but also for m- and v-RNA (Comp. Supplementary Fig. 13 and 14).

In order to draw the general conclusion that the sequence identified at the 5' end of PB1 has an effect on IAV replication, the authors should introduce the two critical mutations (C2319G and U2335G) into an authentic WSN virus by reverse genetics and perform growth kinetics.

(Remarks on code availability)

Reviewer #2

(Remarks to the Author)

It has been reported that the NS2 protein of influenza virus plays a critical role in mediating the nuclear export of vRNP and is involved in the regulation of viral RNA transcription and replication. This manuscript by Swaminath et al., focused on the

study of the association between the viral RNA sequences and the role of NS2 in the regulation of viral RNA replication. The authors previously studied the replication efficiency of defective influenza virus RNA segments with different lengths (Mendes M and Russell AB, PLoS Pathog. 2021). In this study, with the previously established length-variant libraries of PB1 and HA, they attempted to assess the effects of NS2 on the replication efficiencies of these template libraries in a mini-replicon system. Through deletions and random mutagenesis of the genomic RNA, they discovered that certain nucleotides in the terminal promoter and adjacent regions of the genomic RNA are critical for the NS2-dependent viral RNA replication. As a result, the authors claimed: "Efficient genome replication in influenza A virus requires NS2 and sequence beyond the canonical promoter." In general, the data presented in this manuscript do not fully support the authors' conclusion, and the research does not provide valuable insights into the regulation of influenza virus genomic RNA replication.

Major points:

1. It is well known that polymerase processivity is the result of a combination effect of template length and RNA structure stabilities (sequences). As the author stated that different deletions may influence polymerase processivity by varying template RNA structures regardless of template lengths. The inconsistency described in the Figure 1 with different templates and in different systems may also be a result of the different sequences of the template, in addition to the complexity envisaged by the authors.

2. As the authors referenced, the regulatory role of NS2 in promoting viral RNA replication has been widely observed and reported by several papers. Figure 6b in this m/s also confirmed this effect. However, in Figure 2: firstly, the author did not make it clear which viral RNA species (cRNA or vRNA?) was detected in the qPCR. Secondly, the expression of NS2 protein in this figure and in subsequent figures generally reduces the level of viral RNA replication, which is contradictory with the literature and their own primer extension analysis presented in Figure 6b & Supplementary figure 12 and 14. These results may throw doubts to the system they used.

3. Both function and structural studies have clearly demonstrated that the promoter region of the influenza virus genomic RNA is crucial for viral RNA transcription and replication. In this study, the authors performed extensive random mutagenesis screening in the vRNA promoter regions (Fig. 5), which would themselves severely and differentially impact vRNA transcription and replication, e.g. point mutation at position 2, 3, 5 would kill promoter activity. In figure 5, the authors used $\log_2(\text{NS2}/\text{NS2-})$ as readout which could not directly reflect the actual promoter activity. From the value of Y axis in Figure 5, it seems that these key promoter mutations only resulted in the reduction of polymerase activity about 2-30 folds, which do not make sense. Furthermore, since the transfected plasmids were present in the system and could be amplified by qPCR, the author should include a negative control with polymerase omitted in the system to validate the system.

4. In Figure 5b, the authors constructed PB1_177:385-defective RNA library using error-prone PCR and transfected them in the mini-replicon system with or without additional expression of NS2. By using the library, they identified four sites (U20, G28, C2319, and U2335) within and beyond the promoter that are required for NS2-dependent viral RNA replication. However, the solid primer extension analyses in Figure 6b shows that U20A in the PB1_177:385 template did not affect NS2-dependent viral RNA replication whereas this mutation in the full-length PB1 segment significantly affected the NS2-dependent viral RNA replication (PB1_177:385 template). Furthermore, G28, C2319 in the PB1_177:385 template disrupted NS2 effect on viral RNA replication while the two mutations in the full-length PB1 template (PB1_177:385 template) did not affect the NS2 effects. Therefore, three out of the four identified point mutations showed inconsistent results which would weaken the conclusion of the manuscript.

Minor Issues:

1. In Figure 3, the range of length of PB1 (left panel) was not correct. The full length of PB1 is 2.3kb.
2. Lines 250-253: The reference (Staller E et al., Nat Commun., 2024) is missing.
3. Figure 4b, the labels in X axis should be 0.02-0.12kb.

(Remarks on code availability)

Reviewer #3

(Remarks to the Author)

The study by Swaminath et al., provides an interesting and nuanced glimpse into the complexities of IAV genome replication, focusing on the critically overlooked role of NS2. Through a range of mutant libraries of HA and PB1, with which Dr Russell is well versed, the authors explore and describe the importance of conserved regions outside the U12 and U13 for RNA replication. This is an important finding in the field, while the methodology laid out to come to these conclusions may be useful to researcher's interested in other viruses.

I think this is an interesting, well thought out and executed study which conclusively describes the importance of NS2 for viral replication, with the inclusion of adequate caveats and conclusions of the data in the discussion. Other than some minor comments listed below I have no issues or concerns with this manuscript in its current form and would be happy to see it accepted for publication.

Minor comment

1. I feel that Supplementary figure 3 is more useful for the reader than Figure 4D and would advise that these panels be

switched and the text altered accordingly.

2. On line 176 the author's state "both cRNA and vRNA, which spontaneously convert to cDNA." Spontaneously is not the right term to use in this context as this implies they convert completely independent of other factor when in actuality they convert simply in the absence of a primer but still require an RT. Perhaps the term 'non-discriminately' would work better, or something along those lines.

(Remarks on code availability)

Version 1:

Reviewer comments:

Reviewer #1

(Remarks to the Author)

The authors have addressed the issues raised and improved the paper a lot! Well done!

(Remarks on code availability)

Reviewer #2

(Remarks to the Author)

After spending several hours reviewing this revision, I still feel it has not adequately addressed the reviewer's comments in general. Since the major concerns remain unresolved, so I won't delve into details again at length. However, I would like to summarize the primary issues to clarify:

1. Regarding the infection experiment, the authors still used an artificial infection system rather than an authentic WSN virus infection. The results obtained from their single-round infection system are limited to measuring the RNA levels from artificial templates and do not reflect the growth kinetics of virus infection. As a result, the conclusion and the broader biological significance of this study remains uncertain.

2. While the role of NS2 in modulating viral RNA synthesis has been well-established, this study's approach on the segment-specific regulatory effects of NS2 raised concerns. The reliance on artificially truncated and sequence variants of PB1 or HA segment in the mini-replicon system is highly unnatural and complicated and fails to replicate segment-specific viral RNA synthesis under conditions involving eight full-length segments during authentic infection. Furthermore, the effects of NS2 observed in the mini-replicon or the single-round infection system cannot completely rule out the effects of variations in library/template transfection efficiencies, the influence of RNA template structures on polymerase processivity, the effect of NS2 on vRNP nuclear export. To draw definitive conclusions, the identified key nucleotides must be validated in authentic infection system, demonstrating their impact on segment-specific RNA synthesis dynamics and virus growth kinetics.

(Remarks on code availability)

Reviewer #3

(Remarks to the Author)

I'd like to thank the authors for their positive and timely responses and commend them on the additional work they have included which clearly strengthens the overall findings. I am happy with the manuscript at present and have no further issues.

(Remarks on code availability)

Thank you to all three reviewers for your time and attention on our manuscript. We recognize that everyone is quite busy nowadays, and appreciate the effort put in to assess our work. We have updated our manuscript to address concerns you all have raised, and feel that thanks to your critiques we have produced a significantly stronger work.

We address each individual reviewer below.

Reviewer #1 (Remarks to the Author):

In this study, Swaminath and colleagues investigate the impact of the Influenza A virus NEP (formerly known as NS2) on the replication of individual viral genome sequences. They can show for a PB1 fragment that the impact of NEP on its replication depends on a sequence located at the 5' end, which is not part of the core promoter.

It has been known for several years that NEP, in addition to its role in the nuclear export of newly synthesized viral genome segments, is also a crucial co-factor of the viral polymerase. The underlying mechanism is still largely unknown. To investigate the influence of NEP on the replication of individual genome segments, the authors employ polymerase reconstitution assays. They use a short variant of the PB1 segment to identify critical sequence positions for NEP-dependent replication. Verification of these positions in a full-length PB1 segment was insufficiently performed. Furthermore, this reviewer feels that the relevance of the here presented results is limited without an evaluation of the critical positions in the context of an authentic viral infection.

The title of the manuscript is misleading. With the exception of a PB2 variant of 200nt, the presence of NEP results in an overall reduction of replication (Comp. Fig. 2A). Yet the authors claim in their title that efficient genome replication in IAV requires NEP. Further, the authors focus on an individual segment: PB1. Sequence-specific effects were not studied for other segments. This should be reflected in the title.

The authors use a 562nt PB1 fragment (PB1_177:385) in order to identify a critical sequence at the 5' end. Although the authors state that the PB1_177:385 fragment was chosen to increase the dynamic range of their measurements, it is yet very artificial. The authors should repeat the experiment shown in Fig. 4c in the context of full length PB1.

Along these lines, the experiment shown in Figure 6a should be repeated in the context of full length PB1. Further a quantification of the primer extension analysis should not only be conducted for cRNA, but also for m- and v-RNA (Comp. Supplementary Fig. 13 and 14).

In order to draw the general conclusion that the sequence identified at the 5' end of PB1 has an effect on IAV replication, the authors should introduce the two critical mutations (C2319G and U2335G) into an authentic WSN virus by reverse genetics and perform growth kinetics.

We thank the reviewer for their critical assessment of our work. To address each point in turn.

- Most critically, we agree that the sites we identify should be verified in full-length PB1, in the context of infection. We have consistently had issues with stochastic reconstitution of the full-length template (now shown in Supplementary Figure 5), and note that many similar studies have similarly used reduced templates. So while we could not procure convincing data on full-length templates for minimal replication assays (repeating 4c and 6a, as requested), we instead tested all of the NS2-responsive features in full-length PB1 in the context of infection. We found that the full-length template is readily replicated but all of our NS2-responsive identified point mutations (C2319G, U2335G, G28C, and U20A) and the identified deletion (75-125) showed decreased vRNA levels and had a negative effect on

replication in a single-round replication assay. This can be found in the new panel, Figure 6c. Therefore **all** sequences we chose to explore in our reduced template negatively influence replication in the context of infection and full-length template. This I think nicely verifies the biological significance of our findings, and justifies the use of both competition assays (better explained now in the new Supplementary Figure 1) and the inclusion of NS2 to study selection on sequence required for replication in influenza A virus. We thank you a great deal for this suggestion as we feel this greatly improves the significance and reach of our findings.

- We agree that our title should have been more accurate. To address this concern we have updated the title to “Analysis of NS2-dependent effects on influenza PB1 segment extends replication requirements beyond the canonical promoter”. Our original effort was trying to be somewhat agnostic about whether NS2 increased replication of longer templates or decreased those lacking these sequences, however we have now gone ahead and rigorously measured this difference both with the inclusion of our infection experiments (full length in 6c and shortened in 4d, which show exclusion of templates lacking these sequences) as well as measurements of a cellular housekeeping control to provide a more fair comparison within Figure 2 (Supplementary Figure 3). We also agree that while these effects are probably present in more than PB1, we do not actually show data in support of that model. We hope you find our updated title is both more precise as to the work that was actually done as well as the conclusions we draw.
- We have also updated the manuscript with a full measurement of all molecular species in our full-length primer-extension assays. We have updated both the text, and added Supplementary Figure 1 to explain how these assays may differ from molecular competition assays, which make up the bulk of our paper. Critically, while the exact molecular mechanisms through which these mutations lead to exclusion of replication still remains somewhat unclear, all of our identified mutations perturb replication in the context of viral infection when introduced to full-length PB1 template.

We thank you once more for your critical assessment, as now our work can be fairly related to infection without lingering doubts as to its relevance. This is much appreciated and we hope you find our revised manuscript of more broad interest as well as improved in rigor.

Reviewer #2 (Remarks to the Author):

It has been reported that the NS2 protein of influenza virus plays a critical role in mediating the nuclear export of vRNP and is involved in the regulation of viral RNA transcription and replication. This manuscript by Swaminath et al., focused on the study of the association between the viral RNA sequences and the role of NS2 in the regulation of viral RNA replication. The authors previously studied the replication efficiency of defective influenza virus RNA segments with different lengths (Mendes M and Russell AB, PLoS Pathog. 2021). In this study, with the previously established length-variant libraries of PB1 and HA, they attempted to assess the effects of NS2 on the replication efficiencies of these template libraries in a mini-replicon system. Through deletions and random mutagenesis of the genomic RNA, they discovered that certain nucleotides in the terminal promoter and adjacent regions of the genomic RNA are critical for the NS2-dependent viral RNA replication. As a result, the authors claimed: “Efficient genome replication in influenza A virus requires NS2 and sequence beyond the canonical promoter.” In general, the data presented in this manuscript do not fully support the authors’ conclusion, and the research does not provide valuable insights into the regulation of influenza virus genomic RNA replication.

We thank you for your time in assessing our manuscript. We do think there has been some confusion regarding our methods and data we presented. Recognizing that it is our job, as the authors, to communicate our points clearly, we have substantially updated our manuscript to try and address these points of confusion. With these changes, and the addition of infection data in full-length PB1 template, we hope you find the amended manuscript a considerable contribution to the field.

Major points:

1. It is well known that polymerase processivity is the result of a combination effect of template length and RNA structure stabilities (sequences). As the author stated that different deletions may influence polymerase processivity by varying template RNA structures regardless of template lengths. The inconsistency described in the Figure 1 with different templates and in different systems may also be a result of the different sequences of the template, in addition to the complexity envisaged by the authors.

We agree that different templates can be idiosyncratic. We attempted to convey this concept in our original manuscript in lines 70-82, and again in lines 100-102. The point we were hoping to convey is that when we use our length-variable libraries, we sample each length across many different deletions and duplications, so any given deletion junction sequence would be averaged over multiple examples of that length. This is the advantage of our approach. We do not simply show how a particular template replicates, but how multiple templates of that length replicate.

Based on minor concerns #1 and #3, I think we failed to convey how, exactly, these libraries work. While they were published previously, we do think it is good to convey all information to the reader so that there are not these misunderstandings. We have updated our text (new lines 124-130) to be more specific about the nature of our libraries, which were performed to exhaustively determine length and have now been repurposed to identify region/sequence specific effects on replication upon NS2 expression.

2. As the authors referenced, the regulatory role of NS2 in promoting viral RNA replication has been widely observed and reported by several papers. Figure 6b in this m/s also confirmed this effect. However, in Figure 2: firstly, the author did not make it clear which viral RNA species (cRNA or vRNA?) was detected in the qPCR. Secondly, the expression of NS2 protein in this figure and in subsequent figures generally reduces the level of viral RNA replication, which is contradictory with the literature and their own primer extension analysis presented in Figure 6b & Supplementary figure 12 and 14. These results may throw doubts to the system they used.

In our original methods we described using a vRNA-specific primer to generate cDNA for qPCR (original line 310). We agree it is best to be clear about all details, so we have expanded upon this with a discussion in the main text (new lines 87-89).

Regarding the decrease in replication we have a few comments:

- 1) We did actually see a decrease in vRNA in the original Figure 6c (now 6b) for C2319G and U2335G, so we are a little confused regarding the statement that we do not see any decreases.
- 2) With full-length template now fully processed we also see NS2-dependent decreases in vRNA for U20A (Supplementary Figure 15).
- 3) Respectfully, the decreases are not inconsistent with prior literature. For our wild-type (or parental) templates, we see data that are highly consistent with others (increase in cRNA, decrease in mRNA). **It is only for our mutant templates that we describe a decrease in replication.** As this is, to our knowledge, the very first time anyone has altered sequence in the context of NS2-dependent replication there are no contradictions with the literature because there is no literature. If we have erred, and there

are pre-existing experiments with variants as we describe, we definitely would appreciate being corrected and being provided with the specific reference as we do want to place our work within the greater body of literature.

- 4) Regardless, I think there is a misunderstanding about what the primer-extension assays measure relative to the qPCR assays in Figure 6a (as well as other qPCR assays in Figures 2 and 4). All qPCR assays are competition assays (described further in our new Supplementary Figure 1). Primer extension would be challenging in this context as one would need unique sequences to prime against that are relatively close to the 5' end of each template. **The experiment in 6a is a verification of our findings, in the context of a molecular competition, whereas the primer extension assays are measuring steady-state concentrations in the absence of a molecular competitor.** As we elaborate in Supplementary Figure 1, and lines 90-100 in the revised manuscript, the former would be expected to be more sensitive to the phenomena we are most interested in, even if we unfortunately cannot say for certain at what stage of replication each variant is acting.

Still, we accept you may still have some skepticism. We now present Figure 6c, demonstrating all NS2-responsive regions that we identify negatively influence replication in the context of viral infection with a full-length template. We hope you find these data as compelling as we do, and that it assuages any concerns you have about their relevance.

3. Both function and structural studies have clearly demonstrated that the promoter region of the influenza virus genomic RNA is crucial for viral RNA transcription and replication. In this study, the authors performed extensive random mutagenesis screening in the vRNA promoter regions (Fig. 5), which would themselves severely and differentially impact vRNA transcription and replication, e.g. point mutation at position 2, 3, 5 would kill promoter activity. In figure 5, the authors used log₂+NS2/-NS2 as readout which could not directly reflect the actual promoter activity. From the value of Y axis in Figure 5, it seems that these key promoter mutations only resulted in the reduction of polymerase activity about 2-30 folds, which do not make sense. Furthermore, since the transfected plasmids were present in the system and could be amplified by qPCR, the author should include a negative control with polymerase omitted in the system to validate the system.

First, we agree, showing the dynamic range is important. This is now provided in Supplementary Figure 5. We are not certain which aspect of the transfected plasmids you are concerned about, so to explain all:

- 1) Our sequencing of a shortened template involves a gel purification step of shortened product, excluding mRNA from the PB1 expression vector.
- 2) Our sequencing uses a DNase step, excluding plasmid DNA.
- 3) We do expect vRNA produced from the vector to provide a baseline (so this is now addressed in Supplementary Figure 5).
- 4) Our cRNA sequencing would actually exclude even this contamination as it is a 5' RACE.

We find the ~3-5 CT difference observed in Supplementary Figure 5 to be consistent with the strongest effect sizes we observe. A quick note, the effect size in a complex mixture of templates is unlikely to exactly match that observed in single, and we, as others who use such libraries, focus on the directionality and relative magnitude (more is more, less is less, positive remains positive, negative negative). It is unlikely when testing ~2000 different variants simultaneously that data perfectly match in single.

Regardless, we respect that you would like to see more confirmation of the validity of our study. In addition to the full-length infection data, we have re-processed our libraries to emphasize how well we predict the viral promoter (new Supplementary Figure 11). Promoter sites in general are much more stringently selected, and we recover the wild-type nucleotide as the most effective for replication when NS2 is concurrently expressed.

To be clear, we do not use these data to make claims that we have discovered the promoter, but rather that amongst our ~2000 measurements, we correctly identify the canonical promoter as having a high degree of stringency. Recovery of each and every site in the U12/U13 as significant under NS2 expression speaks to the efficacy, replicability, and biological meaning of our analysis. Thank you for asking us to more vigorously verify our results; we now present a more convincing story as to the strength of our data.

To address your other concern. We did search the literature to try and determine which paper refers to the positions you are discussing. We believe you are referring to “Mutational Analysis of the Influenza Virus cRNA Promoter and Identification of Nucleotides Critical for Replication.” *J. Virol.*, 2004. which used an NS-flanked CAT gene as its template. As a quick aside, while ideally it would not matter that the template in that study was (+) sense, a prior effort from Dr. Peter Palese’s group using the same template but (-) sense observed significant mRNA expression in templates with mutations to 3’ positions 1,3, 8 and 9, which have no mRNA in this study. A study from Dr. Ervin Fodor’s group, using a template derived from neuraminidase, similarly found that position 10, 5’ of the vRNA, reduced mRNA expression whereas in this paper it did not.

None of this is to cast any aspersions on 2004 *J. Virol* paper, which is wonderful and a solid contribution to the field. But rather, template choice and experimental conditions (including timing of sampling) appear to strongly set the strength of the observed phenotypes.

It would be quite troubling if a mutation that fully “killed” polymerase activity had an intermediate phenotype in our assays. So we have gone ahead and tested one of these positions, position 5, now in Supplementary Figure 11d, and found that we can detect vRNA replication. In addition to using primers specific for vRNA for cDNA generation, to rigorously test whether any genome replication remained we used a primer pair that excludes **any** RNA from our PB1 expression construct, so the signal observed must come from replicated genomes and not from any other source. We do not test the other positions as the point of our paper is not to try and redefine the canonical promoter, which as this reviewer notes we are predated by significant work, but rather to redefine sequence required for replication upon NS2 expression. Given all of the other metrics by which our libraries succeed, including, most importantly, our verification of sites of interest in infection in the full-length template, we hope you are as satisfied as we are with these explanations and additional data.

4. In Figure 5b, the authors constructed PB1_177:385-defective RNA library using error-prone PCR and transfected them in the mini-replicon system with or without additional expression of NS2. By using the library, they identified four sites (U20, G28, C2319, and U2335) within and beyond the promoter that are required for NS2-dependent viral RNA replication. However, the solid primer extension analyses in Figure 6b shows that U20A in the PB1_177:385 template did not affect NS2-dependent viral RNA replication whereas this mutation in the full-length PB1 segment significantly affected the NS2-dependent viral RNA replication (PB1_177:385 template). Furthermore, G28, C2319 in the PB1_177:385 template disrupted NS2 effect on viral RNA replication while the two mutations in the full-length PB1 template (PB1_177:385 template) did not affect the NS2 effects. Therefore, three out of the four identified point mutations showed inconsistent results which would weaken the conclusion of the manuscript.

We again hope that Supplementary Figure 1 helps in the assessment of the data. The differences between templates likely reflect different kinetic requirements setting steady-state concentrations, also described further in lines 90-100 of the updated manuscript. We include the data for the sake of transparency, but they do not impact our ultimate conclusion; that these sites are critical for replication, and that NS2 expression in minimal replication assays defines them as required. Our competition data in Figure 6a are meant to be the confirmation, as they are the only **kinetic** assays provided. We attempted to explain this in our original manuscript (lines 225-229), but hope the inclusion of the Supplementary Figure 1 schematic helps to make things more clear.

Regardless, we agree ending the story on a bit of a confusing note was not the best idea, although we do now emphasize that all positions appear to perturb RNA generation in some way shape or form upon NS2 expression, even if the exact mechanisms remain unclear (new lines 280-291).

Therefore we again hope that our inclusion of **infection** data using **full-length** PB1 template removes any concern as to the relevance of our findings.

Minor Issues:

1. In Figure 3, the range of length of PB1 (left panel) was not correct. The full length of PB1 is 2.3kb.

As we describe above this seems to be a misunderstanding of our method. We do actually recover longer PB1 segments at sufficient abundance to define measurements with statistical confidence. Lines 123-130 show the range of the recovered lengths in this library.

2. Lines 250-253: The reference (Staller E et al., Nat Commun., 2024) is missing.

We are happy to include this reference in addition to the other ANP32A references we provided (now reference #61, line 321).

3. Figure 4b, the labels in X axis should be 0.02-0.12kb.

Also as described above there seems to be some confusion. We have updated Figure 4b, the caption, as well as the text (at lines 153-160). The X axis is correct, fragment sizes we reliably recover with, and without, the region at around 100nt into the 5' end of the template are 200nt, 400nt, 600nt, 800nt, 1kb, and 1.2kb. When we examine each individual length, and compare those with, and without, this region, with, and without, NS2, we observe the indicated depletion. This indicates it is not one single deletion junction driving our observation, but the specific absence of that region that reduces replication upon NS2 expression. We also hope that the additional color code aids the understanding of this point.

Thank you for assessing our paper. We have added considerable rigor, and tried to improve how we have conveyed our story. We respect some of our findings with NS2 were surprising, but as we have now validated the importance of these positions in infection, as well as provided extensive support for our experimental approach, we hope you find our revised manuscript compelling and a contribution to the field.

Reviewer #3 (Remarks to the Author):

The study by Swaminath et al., provides an interesting and nuanced glimpse into the complexities of IAV genome replication, focusing on the critically overlooked role of NS2. Through a range of mutant libraries of HA and PB1, with which Dr Russell is well versed, the authors explore and describe the importance of conserved regions outside the U12 and U13 for RNA replication. This is an important finding in the field, while the methodology laid out to come to these conclusions may be useful to researcher's interested in other viruses.

I think this is an interesting, well thought out and executed study which conclusively describes the importance of NS2 for viral replication, with the inclusion of adequate caveats and conclusions of the data in the discussion. Other than some minor comments listed below I have no issues or concerns with this manuscript in its current form and would be happy to see it accepted for publication.

Thank you for your kind words. We appreciate the time and effort you spent to review our work, and are glad you found it interesting!

Minor comment

1. I feel that Supplementary figure 3 is more useful for the reader than Figure 4D and would advise that these panels be switched and the text altered accordingly.

We agree. Altered accordingly. We have gone ahead and also added a similar analysis to Figure 5 (moving it up from the supplement). We also have added a considerable analysis as to the rigor of our approach which you might find interesting, now provided as Supplementary Figure 11.

2. On line 176 the author's state "both cRNA and vRNA, which spontaneously convert to cDNA." Spontaneously is not the right term to use in this context as this implies they convert completely independent of other factor when in actuality they convert simply in the absence of a primer but still require an RT. Perhaps the term 'non-discriminately' would work better, or something along those lines.

Quite correct, we apologize for our imprecise language. We have now discussed this more at-length and quantitatively in new lines 87-89 and again at lines 205-207.

Thank you again!

We thank all three reviewers for revisiting our manuscript, and for their insightful reviews. As a final response to reviewers, we would like to textually address the confusion and concerns of reviewer #2.

REVIEWERS' COMMENTS

Reviewer #1 (Remarks to the Author):

The authors have addressed the issues raised and improved the paper a lot! Well done!

Thank you for your kind words, as well as for your critical review.

We appreciate your comments, critiques, and suggestions which we feel greatly improved our paper.

Reviewer #2 (Remarks to the Author):

After spending several hours reviewing this revision, I still feel it has not adequately addressed the reviewer's comments in general. Since the major concerns remain unresolved, so I won't delve into details again at length. However, I would like to summarize the primary issues to clarify:

It still appears that there is confusion regarding the results we have presented. If we are interpreting your comments as you intended, then you seem to be suggesting an experiment that would provide uninterpretable results due to feedback loops and secondary effects that cannot be disentangled. We will elaborate further, point-by-point, below. We are certain there must be significant misunderstandings, so we are happy to address these remaining concerns textually.

1. Regarding the infection experiment, the authors still used an artificial infection system rather than an authentic WSN virus infection. The results obtained from their single-round infection system are limited to measuring the RNA levels from artificial templates and do not reflect the growth kinetics of virus infection. As a result, the conclusion and the broader biological significance of this study remains uncertain.

We acknowledge that we could have more clearly conveyed our experimental setup and rationale.

In regards to Figure 6c, a WSN single round infection, the artificial aspects include ectopically expressed PB1 and our barcoded 1600nt variant of PB1 that we used as a competitor (we used this competitor due to the concerns we outlined extensively in Supplementary Figure 1). The PB1 vRNA segment we are measuring, and which we introduce variants of interest, only bears one additional mutation, the removal of the start codon. We updated the methods (lines 320-333 and 447-452) to be as clear as we can regarding this experiment.

For these results to lack biological relevance, we would need to presume that these sequences: have an effect during our viral replication experiments with full-length template; have an effect in minimal replication experiments with a minimal template; and, lastly are unusually conserved throughout PB1 sequences and exhibit a high preference for wild-type sequence during replication.

What you appear to be asking is putting these mutations into otherwise wild-type viruses. This should not be performed, as it would produce uninterpretable results.

To explain further:

If introduced, these mutations we chose would create a feedback on PB1 expression, particularly with respect to the polyU mutant on mRNA levels and the chosen deletion altering the coding sequence. These *cis* features of the sequence cannot be disambiguated from the *trans* effects on PB1 expression and function. Additionally, "the growth kinetics of virus infection" would include export and packaging effects which would further confound results. Unfortunately, we must conclude that there is no way to cleanly interpret the results of the authentic infection you seem to desire. We have updated the text to point out these concerns (lines 320-333).

2. While the role of NS2 in modulating viral RNA synthesis has been well-established, this study's approach on the segment-specific regulatory effects of NS2 raised concerns. The reliance on artificially truncated and sequence variants of PB1 or HA segment in the mini-replicon system is highly unnatural and complicated and fails to replicate segment-specific viral RNA synthesis under conditions involving eight full-length segments during authentic infection.

To begin addressing point 2, we have two initial issues we would like to address. First, and as described above, we were able to perform an infection in Figure 6c that confirmed the replication behaviour of an NS2-sensitive region and specific nucleotides. We assume that your concern is due to confusion regarding this experiment.

Secondly, we think it would be inappropriate, and unfair to the field as a whole, if we did not defend the use of mini-replicons at least a little. While mini-replicons are artificial systems, and do not recapitulate full infection dynamics, they are still very useful. We absolutely agree that confirmation of results from such systems are important (hence our experiment in 6c), or, when confirmation is impossible, in the very least accepting their limitations.

However, we feel we must point out the entirety of the original efforts to map of the influenza promoter, as well as the work defining NS2 as promoting cRNA production, are all based on work from mini-replicons. This makes sense given the limitation that many promoter mutations likely cannot be rescued at all. As you, the reviewer, extensively rely on such work in your first critique of our manuscript, the strong language here would seem to suggest that you must feel the entire field is in error. What then confuses us is how you seem to simultaneously rely on that literature to critique us, while at the same time seemingly disparaging it here.

We will address these next points individually.

Furthermore, the effects of NS2 observed in the mini-replicon or the single-round infection system cannot completely rule out the effects of variations in library/template transfection efficiencies,

Regarding our minimal replication assays, we wholeheartedly agree that transfection efficiency is a concern that researchers should be aware of. This concern is why we include internal competitors, which is the gold standard for regressing against such variation.

With concerns to our libraries:

- All libraries are "one-pot reactions," meaning all templates are subjected to identical conditions
- As is the standard in the field for every submission, we included reproducibility metrics (now Supplementary Figures 4, 9, and 10). Our libraries are from true biological replicates, including library generation, which we explicitly stated in the methods. This information was made available during our first submission.
- Lastly, in our second submission, we demonstrate the recovery of wild-type sequence in the canonical promoter, indicating our diverse selection recapitulated prior knowledge, increasing confidence in our methods and conclusions about sites outside of what has been considered the canonical promoter.

As for the single-round infection experiments, these only use virus supernatant. Transfection was only used in the rescue of viral supernatant, not the infection experiment itself, and, as indicated in the methods, we use qPCR to correct for the initial inoculum which we have tried to make more clear with additional edits.

the influence of RNA template structures on polymerase processivity,

We are not quite certain what the concern is, as conserved RNA structures could absolutely be a mechanism through which influenza adjusts replication of any given segment. Indeed we agree that RNA template structure on polymerase processivity is important, and is likely a mechanism through which influenza can fine-tune replication as stated in our first submission and remains in lines 63-64 and 89-91.

However, relevant to this manuscript, **we find that processivity induced by NS2 is an unlikely explanation** for our observed results. We have added some additional text at lines 137-139 to further explain our conclusions.

If our results were best explained by a model where increased processivity is the sole driver, we would anticipate the following results which are contradicted by our experimental data:

- Increased processivity should uniformly increase replication of longer templates.
 - This is contradicted by both Figures 2, and Supplementary Figure 6
- During infection, where NS2 is present, RNA structure should be irrelevant and our phenotypes should be absent.
 - This is contradicted by Figure 6c, where we see our variants are at a competitive disadvantage.

the effect of NS2 on vRNP nuclear export.

Export requires M1 and NS2/NEP and is unlikely to be relevant for the minimal replicon assays, which exclude M1. Only during our viral infection should nuclear export be a concern; however, since the scope of our paper concerns replication and we do not measure packaging or downstream steps that are heavily relevant on export we are not sure why this issue is raised here. Export would be a relevant effect to the experiment you seem to propose in point #1; however we have already adequately addressed the limitations of such experiments.

To draw definitive conclusions, the identified key nucleotides must be validated in authentic infection system, demonstrating their impact on segment-specific RNA synthesis dynamics and virus growth kinetics.

Addressed above.

Thank you again for the several hours you spent on this revision. We acknowledge that we could have more clearly expressed our experimental setup, rationale, and conclusions in our initial revision. We hope that our updated text addresses your past and present concerns.

Reviewer #3 (Remarks to the Author):

I'd like to thank the authors for their positive and timely responses and commend them on the additional work they have included which clearly strengthens the overall findings. I am happy with the manuscript at present and have no further issues.

Thank you for your very positive initial review and helpful suggestions.

Thank you as well for going over our revision, due to the diligence of yourself and the other reviewers we have been able to generate a much strengthened manuscript.